# Matérn Gaussian processes on Riemannian manifolds

**Viacheslav Borovitskiy**[*1,4] **Alexander Terenin**[*2] **Peter Mostowsky**[*1] **Marc Peter Deisenroth**[3]

[1]St. Petersburg State University     [2]Imperial College London     [3]University College London
[4]St. Petersburg Department of Steklov Mathematical Institute of Russian Academy of Sciences

## Abstract

Gaussian processes are an effective model class for learning unknown functions, particularly in settings where accurately representing predictive uncertainty is of key importance. Motivated by applications in the physical sciences, the widely-used Matérn class of Gaussian processes has recently been generalized to model functions whose domains are Riemannian manifolds, by re-expressing said processes as solutions of stochastic partial differential equations. In this work, we propose techniques for computing the kernels of these processes on compact Riemannian manifolds via spectral theory of the Laplace–Beltrami operator in a fully constructive manner, thereby allowing them to be trained via standard scalable techniques such as inducing point methods. We also extend the generalization from the Matérn to the widely-used squared exponential Gaussian process. By allowing Riemannian Matérn Gaussian processes to be trained using well-understood techniques, our work enables their use in mini-batch, online, and non-conjugate settings, and makes them more accessible to machine learning practitioners.

## 1 Introduction

Gaussian processes (GPs) are a widely-used class of models for learning an unknown function within a Bayesian framework. They are particularly attractive for use within decision-making systems, e.g. in Bayesian optimization [36] and reinforcement learning [10, 11], where well-calibrated uncertainty is crucial for enabling the system to balance trade-offs, such as exploration and exploitation.

A GP is specified through its mean and covariance kernel. The Matérn family is a widely-used class of kernels, often favored in Bayesian optimization due to its ability to specify smoothness of the GP by controlling differentiability of its sample paths. Throughout this work, we view the widely-used squared exponential kernel as a Matérn kernel with infinite smoothness.

Motivated by applications areas such as robotics [4, 22] and climate science [5], recent work has sought to generalize a number of machine learning algorithms from the vector space to the manifold setting. This allows one to work with data that lives on spheres, cylinders, and tori, for example. To define such a GP, one needs to define a positive semi-definite kernel on those spaces.

In the Riemannian setting, as a simple candidate generalization for the Matérn or squared exponential kernel, one can consider replacing Euclidean distance in the formula with the Riemannian geodesic distance. Unfortunately, this approach leads to ill-defined kernels in many cases of interest [14].

An alternative approach was recently proposed by Lindgren et al. [25], who adopt a perspective introduced in the pioneering work of Whittle [42] and define a Matérn GP to be the solution of

Code available at HTTPS://GITHUB.COM/SPBU-MATH-CS/RIEMANNIAN-GAUSSIAN-PROCESSES and HTTPS://GITHUB.COM/ATERENIN/SPARSEGAUSSIANPROCESSES.JL.

a certain stochastic partial differential equation (SPDE) driven by white noise. This approach generalizes naturally to the Riemannian setting, but is cumbersome to work with in practice because it entails solving the SPDE numerically. In particular, setting up an accurate finite element solver can become an involved process, especially for certain smoothness values [2, 3]. This also prevents one from easily incorporating recent advances in scalable GPs, such as sparse inducing point methods [21, 41], into the framework. This in turn impedes one from easily employing mini-batch training, online training, non-Gaussian likelihoods, or incorporating GPs as differentiable components within larger models.

In this work, we extend Matérn GPs to the Riemannian setting in a fully constructive manner, by introducing Riemannian analogues of the standard technical tools one uses when working with GPs in Euclidean spaces. To achieve this, we first study the special case of the $d$-dimensional torus $\mathbb{T}^d$. Using ideas from abstract harmonic analysis, we view GPs on the torus as periodic GPs on $\mathbb{R}^d$, and derive expressions for the kernel and spectral measure of a Matérn GP in this case.

Building on this intuition, we generalize the preceding ideas to general compact Riemannian manifolds without boundary. Using insights from harmonic analysis induced by the Laplace–Beltrami operator, we develop techniques for computing the kernel and generalized spectral measure of a Matérn GP in this setting. These expressions enable computations via standard GP approaches, such as Fourier feature or sparse variational methods, thereby allowing practitioners to easily deploy familiar techniques in the Riemannian setting. We conclude by showcasing how to employ the proposed techniques through a set of examples.

## 2 Gaussian processes

Let $X$ be a set, and let $f : X \to \mathbb{R}$ be a random function. We say that $f \sim \mathrm{GP}(\mu, k)$ if, for any $n$ and any finite set of points $\boldsymbol{x} \in X^n$, the random vector $\boldsymbol{f} = f(\boldsymbol{x})$ is multivariate Gaussian with prior mean vector $\boldsymbol{\mu} = \mu(\boldsymbol{x})$ and covariance matrix $\mathbf{K}_{\boldsymbol{xx}} = k(\boldsymbol{x}, \boldsymbol{x})$. We henceforth, without loss of generality, set the mean function to be zero.

Given a set of training observations $(x_i, y_i)$, we let $y_i = f(x_i) + \varepsilon_i$ with $\varepsilon_i \sim \mathrm{N}(0, \sigma^2)$. Under the prior $f \sim \mathrm{GP}(0, k)$ the posterior distribution $f \mid \boldsymbol{y}$ is another GP, with mean and covariance

$$\mathbb{E}(f \mid \boldsymbol{y}) = \mathbf{K}_{(\cdot)\boldsymbol{x}}(\mathbf{K}_{\boldsymbol{xx}} + \sigma^2 \mathbf{I})^{-1}\boldsymbol{y} \quad \mathrm{Cov}(f \mid \boldsymbol{y}) = \mathbf{K}_{(\cdot,\cdot)} - \mathbf{K}_{(\cdot)\boldsymbol{x}}(\mathbf{K}_{\boldsymbol{xx}} + \sigma^2 \mathbf{I})^{-1}\mathbf{K}_{\boldsymbol{x}(\cdot)} \quad (1)$$

where $(\cdot)$ denotes an arbitrary set of test locations. The posterior can also be written

$$(f \mid \boldsymbol{y})(\cdot) = f(\cdot) + \mathbf{K}_{(\cdot)\boldsymbol{x}}(\mathbf{K}_{\boldsymbol{xx}} + \sigma^2 \mathbf{I})^{-1}(\boldsymbol{y} - f(\boldsymbol{x}) - \boldsymbol{\varepsilon}) \quad (2)$$

where equality holds in distribution [43]. This expression allows one to sample from the posterior by first sampling from the prior, and transforming the resulting draws into posterior samples.

On $X = \mathbb{R}^d$, one popular choice of kernel is the *Matérn* family with parameters $\sigma^2, \kappa, \nu$, defined as

$$k_\nu(x, x') = \sigma^2 \frac{2^{1-\nu}}{\Gamma(\nu)} \left( \sqrt{2\nu} \frac{\|x - x'\|}{\kappa} \right)^\nu K_\nu \left( \sqrt{2\nu} \frac{\|x - x'\|}{\kappa} \right) \quad (3)$$

where $K_\nu$ is the modified Bessel function of the second kind [17]. The parameters of this kernel have a natural interpretation: $\sigma^2$ directly controls variability of the GP, $\kappa$ directly controls the degree of dependence between nearby data points, and $\nu$ directly controls mean-square differentiability of the GP [29]. As $\nu \to \infty$, the Matérn kernel converges to the widely-used squared exponential kernel

$$k_\infty(x, x') = \sigma^2 \exp\left( -\frac{\|x - x'\|^2}{2\kappa^2} \right) \quad (4)$$

which induces an infinitely mean-square differentiable GP.

For a bivariate function $k : X \times X \to \mathbb{R}$ to be a kernel, it must be *positive semi-definite*, in the sense that for any $n$ and any $\boldsymbol{x} \in X^n$, the kernel matrix $\mathbf{K}_{\boldsymbol{xx}}$ is positive semi-definite. For $X = \mathbb{R}^d$, a translation-invariant kernel $k(x, x') = k(x - x')$ is called *stationary*, and can be characterized via Bochner's Theorem. This result states that a translation-invariant bivariate function is positive definite if and only if it is the Fourier transform of a finite non-negative measure $\rho$, termed the *spectral measure*. This measure is an important technical tool for constructing kernels [29], and for practical approximations such as *Fourier feature* basis expansions [20, 28].

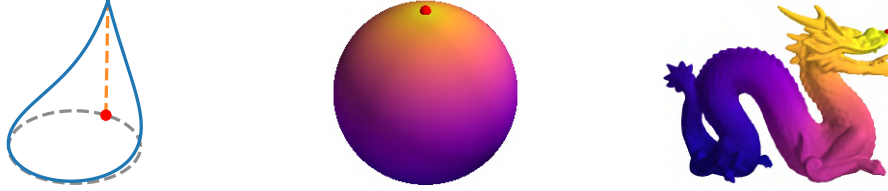

Figure 1: The Matérn kernel $k_{1/2}(x, \cdot)$, defined on a circle, sphere and dragon. The point $x$ is marked with a red dot. The height of the solid line and color, respectively, give the value of the kernel.

## 2.1  A no-go theorem for kernels on manifolds

We are interested in generalizing the Matérn family from the vector space setting to a compact Riemannian manifold $(M, g)$ such as the sphere or torus. One might hope to achieve this by replacing Euclidean norms with the geodesic distances in (3) and (4). In the latter case, this amounts to defining

$$k(x, x') = \sigma^2 \exp\left(-\frac{d_g(x, x')^2}{2\kappa^2}\right) \qquad (5)$$

where $d_g$ is the geodesic distance with respect to $g$ on $M$. Unfortunately, one can prove this is not generally a well-defined kernel.

**Theorem 1.** *Let $(M, g)$ be a complete, smooth Riemannian manifold without boundary, with associated geodesic distance $d_g$. If the geodesic squared exponential kernel (5) is positive semi-definite for all $\kappa > 0$, then $M$ is isometric to a Euclidean space.*

*Proof.*  Feragen et al. [14, Theorem 2]. $\qquad\qquad\qquad\qquad\qquad\qquad\qquad\qquad\qquad\qquad$ □

Since Euclidean space is not compact, this immediately implies that (5) is not a well-defined kernel on any compact Riemannian manifold without boundary. We therefore call (5) and its finite-smoothness analogues the *naïve generalization*.

In spite of this issue, the naïve generalization is usually still positive semi-definite for some $\kappa$, and it has been used in a number of applied areas [22]. Feragen and Hauberg [13] proposed a number of open problems arising from these issues. In Section 3, we show that, on the torus, the naïve generalization is *locally correct* in a sense made precise in the sequel. We now turn to an alternative approach, which gives well-defined kernels in the general case.

## 2.2  Stochastic partial differential equations

Whittle [42] has shown that Matérn GPs on $X = \mathbb{R}^d$ satisfy the stochastic partial differential equation

$$\left(\frac{2\nu}{\kappa^2} - \Delta\right)^{\frac{\nu}{2} + \frac{d}{4}} f = \mathcal{W} \qquad (6)$$

for $\nu < \infty$, where $\Delta$ is the Laplacian and $\mathcal{W}$ is Gaussian white noise re-normalized by a certain constant. One can show using the same argument that the limiting squared exponential GP satisfies

$$e^{-\frac{\kappa^2}{4}\Delta} f = \mathcal{W} \qquad (7)$$

where $e^{-\frac{\kappa^2}{4}\Delta}$ is the (rescaled) heat semigroup [12, 18]. This viewpoint on GPs has recently been reintroduced in the statistics literature by Lindgren et al. [25], and a number of authors, including Särkkä et al. [33] and Simpson et al. [35], have used it to develop computational techniques, notably in the popular INLA package [32].

One advantage of the SPDE definition is that generalizing it to the Riemannian setting is straightforward: one simply replaces $\Delta$ with the Beltrami Laplacian and $\mathcal{W}$ with the canonical white noise process with respect to the Riemannian volume measure. The kernels of these GPs, computed in the sequel, are illustrated in Figure 1. Unfortunately, the SPDE definition is somewhat non-constructive: it is not immediately clear how to compute the kernel, and even less clear how to generalize familiar tools to this setting. In practice, this restricts one to working with PDE-theoretic discretization

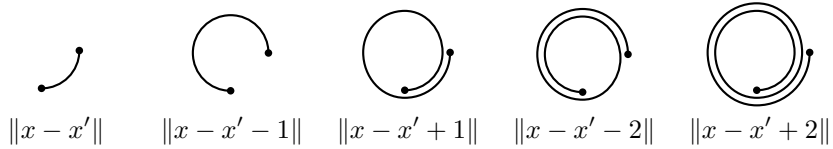

$$\|x - x'\| \qquad \|x - x' - 1\| \qquad \|x - x' + 1\| \qquad \|x - x' - 2\| \qquad \|x - x' + 2\|$$

Figure 2: The distances being considered in definitions (9) and (10).

techniques, such as Galerkin finite element methods, the efficiency of which depend heavily on the smoothness of $f$, and which can require significant hand-tuning to ensure accuracy. It also precludes one from working in non-conjugate settings, such as classification, or from using recently-proposed techniques for scalable GPs via sparse inducing point methods [20, 21, 41], as they require one to either be able to compute the kernel point-wise, or compute the spectral measure, or both.

### 2.3 State of affairs and contribution

In this work, our aim is to generalize the standard theoretical tools available for GPs on $\mathbb{R}^d$ to the Riemannian setting. Our strategy is to first study the problem for the special case of a $d$-dimensional torus. Here, we provide expressions for the kernel of a Matérn GP in the sense of Whittle [42] via *periodic summation*, which yields a series whose first term is the naïve generalization. Building on this intuition, we develop a framework using Laplace–Beltrami eigenfunctions that allows us to provide expressions for the kernel and generalized spectral measure of a Matérn GP on a general compact Riemannian manifold without boundary. The framework is fully constructive and compatible with sparse GP techniques for scalable training.

A number of closely related ideas, beyond those described in the preceding sections, have been considered in various literatures. Solin and Särkkä [38] used ideas based on spectral theory of the Laplace–Beltrami operator to approximate stationary covariance functions on bounded domains of Euclidean spaces. These ideas were applied, for instance, to model ambient magnetic fields using Gaussian processes by Solin et al. [37]. An analog of the expression we provide in equation (18) for the Riemannian Matérn kernel was concurrently proposed as a practical GP model by Coveney et al. [8]—in this work, we *derive* said expression from the SPDE formulation of Matérn GPs. Finally, the Riemannian squared exponential kernel, also sometimes called the heat or diffusion kernel, has been studied by Gao et al. [15]. We connect these ideas with stochastic partial differential equations.

In this work, we concentrate on Gaussian processes $f : M \to \mathbb{R}$ whose *domain* is a Riemannian manifold. We do not study models $f : \mathbb{R} \to M$ where the *range* is a Riemannian manifold—this setting is explored by Mallasto and Feragen [27].

## 3 A first example: the $d$-dimensional torus

To begin our analysis and build intuition, we study the $d$-dimensional torus $\mathbb{T}^d$, which is defined as the product manifold $\mathbb{T}^d = \mathbb{S}^1 \times ... \times \mathbb{S}^1$ where $\mathbb{S}^1$ denotes a unit circle[2]. Since functions on a circle can be thought of as periodic functions on $\mathbb{R}$, and similarly for $\mathbb{T}^d$ and $\mathbb{R}^d$, defining a kernel on a torus is equivalent to defining a periodic kernel. For a general function $f : \mathbb{R}^d \to \mathbb{R}$, one can transform it into a function $g : \mathbb{T}^d \to \mathbb{R}$ by *periodic summation*

$$g(x_1, ..., x_d) = \sum_{n \in \mathbb{Z}^d} f(x_1 + n_1, ..., x_d + n_d) \tag{8}$$

where $x_j \in [0, 1)$ is identified with the angle $2\pi x_j$ and the point $\exp(2\pi i x_j) \in \mathbb{S}^1$. Define addition of two points in $\mathbb{S}^1$ by the addition of said numbers modulo 1, and define addition in $\mathbb{T}^d$ component-wise.

Periodic summation preserves positive-definiteness, since it preserves positivity of the Fourier transform, which by Bochner's theorem is equivalent to positive-definiteness—see Schölkopf and

Smola [34, Section 4.4.4] for a formal proof. This gives an easy way to construct positive-definite kernels on $\mathbb{T}^d$. In particular, we can generalize Matérn and squared exponential GPs from $\mathbb{R}^d$ to $\mathbb{T}^d$ by defining

$$k_\nu(x, x') = \sum_{n \in \mathbb{Z}^d} \frac{\sigma^2 2^{1-\nu}}{C'_\nu \Gamma(\nu)} \left( \sqrt{2\nu} \frac{\|x - x' + n\|}{\kappa} \right)^\nu K_\nu \left( \sqrt{2\nu} \frac{\|x - x' + n\|}{\kappa} \right) \qquad (9)$$

where $C'_{(\cdot)}$ is a constant given in Appendix B to ensure $k_{(\cdot)}(x, x) = \sigma^2$, and

$$k_\infty(x, x') = \sum_{n \in \mathbb{Z}^d} \frac{\sigma^2}{C'_\infty} \exp\left( -\frac{\|x - x' + n\|^2}{2\kappa^2} \right) \qquad (10)$$

respectively. We prove that these are the covariance kernels of the SPDEs introduced previously.

**Proposition 2.** *The Matérn (squared exponential) kernel $k$ in* (9) *(resp.* (10)*) is the covariance kernel of the Matérn (resp. squared exponential) Gaussian process in the sense of Whittle [42].*

*Proof.* Appendix C. $\qquad\qquad\square$

This result offers an intuitive explanation for *why* the naïve generalization based on the geodesic distance might fail to be positive semi-definite on non-Euclidean spaces for all length scales, yet work well for smaller length scales: on $\mathbb{T}^d$, it is *locally correct* in the sense that it is equal to the first term in the periodic summation (9). To obtain the full generalization, one needs to take into account not just geodesic paths, but geodesic-like paths which include loops around the space—a Matérn GP incorporates global topological structure of its domain. For the circle, these are visualized in Figure 2. For spaces where this structure is even more elaborate, definitions based purely on geodesic distances may not suffice to ensure positive semi-definiteness or good numerical behavior. We conclude by presenting a number of practical formulas for Matérn kernels on the circle.

**Example 3** (Circle). *Take $M = \mathbb{S}^1$. For $\nu = \infty$, the kernel and spectral measure are*

$$k_\infty(x, x') = \frac{\sigma^2}{C_\infty} \vartheta_3(\pi(x - x'), \exp(-2\pi^2 \kappa^2)) \qquad \rho_\infty(n) = \frac{\sigma^2}{C_\infty} \exp(-2\pi^2 \kappa^2 n^2) \qquad (11)$$

*where $n \in \mathbb{Z}$, $\vartheta_3(\cdot, \cdot)$ is the third Jacobi theta function [1, equation 16.27.3], and $C_\infty = \vartheta_3(0, \exp(-2\pi^2 \kappa^2))$. This kernel is normalized to have variance $\sigma^2$.*

**Example 4** (Circle). *Take $M = \mathbb{S}^1$. For $\nu = 1/2$, the kernel and spectral measure are*

$$k_{1/2}(x, x') = \frac{\sigma^2}{C_{1/2}} \cosh\left( \frac{|x - x'| - 1/2}{\kappa} \right) \quad \rho_{1/2}(n) = \frac{2\sigma^2 \sinh(1/2\kappa)}{C_{1/2}\kappa} \left( \frac{1}{\kappa^2} + 4\pi^2 n^2 \right)^{-1} \quad (12)$$

*where $C_{1/2} = \cosh(1/2\kappa)$. This kernel is normalized to have variance $\sigma^2$.*

A derivation and more general formula, valid for $\nu = 1/2 + n$, $n \in \mathbb{N}$, can be found in Appendix B. Note that these spectral measures are *discrete*, as the Laplace–Beltrami operator has discrete spectrum. Finally, we give the Fourier feature approximation [20, 28] of the GP prior on $\mathbb{T}^1 = \mathbb{S}^1$, which is

$$f(x) \approx \sum_{n=-N}^N \sqrt{\rho_\nu(n)} \big( w_{n,1} \cos(2\pi nx) + w_{n,2} \sin(2\pi nx) \big) \qquad w_{n,j} \sim \mathrm{N}(0, 1). \qquad (13)$$

We have defined Matérn and squared exponential GPs on $\mathbb{T}^d$ and given expressions for the kernel, spectral measure, and Fourier features on $\mathbb{T}^1$. With sharpened intuition, we now study the general case.

## 4  Compact Riemannian manifolds

The arguments used in the preceding section are, at their core, based on ideas from abstract harmonic analysis connecting $\mathbb{R}^d$, $\mathbb{T}^d$, and $\mathbb{Z}^d$ as topological groups. This connection relies on the algebraic structure of groups, which does not exist on a general Riemannian manifold. As a result, different notions are needed to establish a suitable framework.

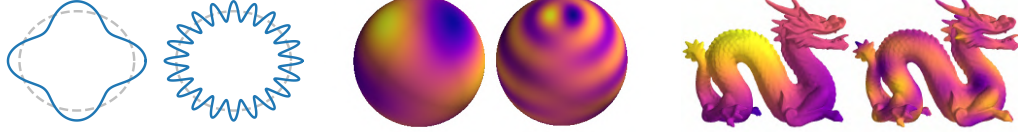

Figure 3: Examples of eigenfunctions of Laplace–Beltrami operator on a circle, sphere, and dragon. For the circle, the value of the eigenfunction is given by the (signed) distance between the solid line and dashed unit circle. For the sphere and dragon, the value of the eigenfunction is given by the color.

Let $(M, g)$ be a compact Riemannian manifold without boundary, and let $\Delta_g$ be the Laplace–Beltrami operator. Our aim is to compute the covariance kernel of the Gaussian processes solving the SPDEs (6) and (7) in this setting. Mathematically, this amounts to introducing an appropriate formalism so that one can calculate the desired expressions using spectral theory. We do this in a fully rigorous manner in Appendix D, while here we present the main ideas and results.

First, we discuss how the operators on the left-hand side of SPDEs (6) and (7) are defined. By compactness of $M$, $-\Delta_g$ admits a countable number of eigenvalues, which are non-negative and can be ordered to form a non-decreasing sequence with $\lambda_n \to \infty$ for $n \to \infty$. Moreover, the corresponding eigenfunctions form an orthonormal basis $\{f_n\}_{n \in \mathbb{Z}_+}$ of $L^2(M)$, and $-\Delta_g$ admits the representation

$$-\Delta_g f = \sum_{n=0}^{\infty} \lambda_n \langle f, f_n \rangle f_n \tag{14}$$

which is termed the *Sturm–Liouville decomposition* [6, 7]. This allows one to define the operators $\Phi(-\Delta_g)$ for a function $\Phi : [0, \infty) \to \mathbb{R}$, by replacing $\lambda_n$ with $\Phi(\lambda_n)$ in (14), and specifying appropriate function spaces as domain and range to ensure convergence of the series in a suitable sense. This idea is called *functional calculus* for the operator $-\Delta_g$. Using it, we define

$$\left(\frac{2\nu}{\kappa^2} - \Delta_g\right)^{\frac{\nu}{2}+\frac{d}{4}} f = \sum_{n=0}^{\infty} \left(\frac{2\nu}{\kappa^2} + \lambda_n\right)^{\frac{\nu}{2}+\frac{d}{4}} \langle f, f_n \rangle f_n \tag{15}$$

$$e^{-\frac{\kappa^2}{4}\Delta_g} f = \sum_{n=0}^{\infty} e^{\frac{\kappa^2 \lambda_n}{4}} \langle f, f_n \rangle f_n. \tag{16}$$

Figure 3 illustrates the eigenfunctions $f_n$. Note that when $M = \mathbb{T}^d$, the orthonormal basis $\{f_n\}_{n \in \mathbb{Z}_+}$ consists of sines and cosines, and thus the corresponding functional calculus is defined in terms of standard Fourier series. This also agrees with the usual way of defining such operators in the Euclidean case using the Fourier transform.

Next, we proceed to define the remaining parts of the SPDEs. The theory of stochastic elliptic equations described in Lototsky and Rozovsky [26] gives an appropriate notion of white noise $\mathcal{W}$ for our setting, as well as a way to uniquely solve SPDEs of the form $\mathcal{L}f = \mathcal{W}$, where $\mathcal{L}$ is a bounded linear bijection between a pair of Hilbert spaces. We show that the operators

$$\left(\frac{2\nu}{\kappa^2} - \Delta_g\right)^{\frac{\nu}{2}+\frac{d}{4}} : H^{\nu+\frac{d}{2}}(M) \to L^2(M) \qquad e^{\frac{\kappa^2}{4}\Delta_g} : \mathcal{H}^{\frac{\kappa^2}{2}}(M) \to L^2(M) \tag{17}$$

are bounded and invertible, where $H^s(M)$ are appropriately defined Sobolev spaces on the manifold, and $\mathcal{H}^s(M)$ are the *diffusion spaces* studied by De Vito et al. [9].

We prove that the solutions of our SPDEs in the sense of Lototsky and Rozovsky [26] are Gaussian processes with kernels equal to the reproducing kernels of the spaces $H^{\nu+d/2}(M)$ and $\mathcal{H}^{\kappa^2/2}(M)$, which are given by De Vito et al. [9]. Summarizing, we get the following.

**Theorem 5.** *Let $\lambda_n$ be eigenvalues of $-\Delta_g$, and let $f_n$ be their respective eigenfunctions. The kernels of the Matérn and squared exponential GPs on $M$ in the sense of Whittle [42] are given by*

$$k_\nu(x, x') = \frac{\sigma^2}{C_\nu} \sum_{n=0}^\infty \left( \frac{2\nu}{\kappa^2} + \lambda_n \right)^{-\nu-\frac{d}{2}} f_n(x) f_n(x') \tag{18}$$

$$k_\infty(x, x') = \frac{\sigma^2}{C_\infty} \sum_{n=0}^\infty e^{-\frac{\kappa^2}{2}\lambda_n} f_n(x) f_n(x') \tag{19}$$

*where $C_{(\cdot)}$ are normalizing constants chosen so that the average variance[3] over the manifold satisfies $\mathrm{vol}_g(M)^{-1} \int_X k_{(\cdot)}(x, x)\mathrm{d}x = \sigma^2$.*

*Proof.* Appendix D. □

Our attention now turns to the spectral measure. In the Euclidean case, the spectral measure, assuming sufficient regularity, is absolutely continuous—its Lebesgue density is given by the Fourier transform of the kernel. In the case of $\mathbb{T}^d$, the spectral measure is discrete—its density with respect to the counting measure is given by the Fourier coefficients of the kernel. Like in the case of the torus, for a compact Riemannian manifold the spectral measure is discrete—its density with respect to the counting measure is given by the generalized Fourier coefficients of the kernel with respect to the orthonormal basis $f_n(x)f_{n'}(x')$ on $L^2(M \times M)$. For Matérn and square exponential GPs, these are

$$\rho_\nu(n) = \frac{\sigma^2}{C_\nu} \left( \frac{2\nu}{\kappa^2} + \lambda_n \right)^{-\nu-\frac{d}{2}} \qquad \rho_\infty(n) = \frac{\sigma^2}{C_\infty} \exp\left( -\frac{\kappa^2}{2}\lambda_n \right) \qquad n \in \mathbb{N}. \tag{20}$$

This allows one to recover most tools used in spectral theory of GPs. In particular, one can construct a regular Fourier feature approximation of the GPs by taking the top-$N$ eigenvalues, and writing

$$f(x) \approx \sum_{n=0}^{N-1} \sqrt{\rho(n)} w_n f_n(x) \qquad w_n \sim \mathrm{N}(0, 1). \tag{21}$$

Other kinds of Fourier feature approximations, such as random Fourier features, are also possible. We now illustrate an example in which these expressions simplify.

**Example 6** (Sphere). *Take $M = \mathbb{S}^d$ to be the $d$-dimensional sphere. Then we have*

$$k_\nu(x, x') = \sum_{n=0}^\infty c_{n,d}\, \rho_\nu(n)\, \mathcal{C}_n^{(d-1)/2}\Big( \cos\big(d_g(x, x')\big) \Big) \tag{22}$$

*where $c_{n,d}$ are constants given in Appendix B, $\mathcal{C}_n^{(\cdot)}$ are the Gegenbauer polynomials, $d_g$ is the geodesic distance, and $\rho_\nu(n)$ can be expressed explicitly in terms of $\lambda_n = n(n + d - 1)$ using (20). See Appendix B for details on the corresponding Fourier feature approximation.*

A derivation with further details can be found in Appendix B. Similar expressions are available for many other manifolds, where the Laplace–Beltrami eigenvalues and eigenfunctions are known.

## 4.1 Summary

We conclude by summarizing the presented method of computing the kernel of Riemannian Matérn Gaussian processes defined by SPDEs. The key steps are as follows.

1. Obtain the Laplace–Beltrami eigenpairs for the given manifold, either analytically or numerically. This step needs to be performed once in advance.
2. Approximate the kernel using a finite truncation of the infinite sums (18) or (19).

This kernel approximation can be evaluated pointwise at any locations, fits straightforwardly into modern automatic differentiation frameworks, and is simple to work with. The resulting truncation error will depend on the smoothness parameter $\nu$, dimension $d$, and eigenvalue growth rate, which is quantified by Weyl's law [44]. For $\nu < \infty$ convergence will be polynomial, and for $\nu = \infty$ it will be exponential. If $\sigma^2$ is trainable, the constant $C_\nu$ which normalizes the kernel by its average variance can generally be disregarded. If Fourier feature approximations of the prior are needed, for instance, to apply the pathwise sampling technique of Wilson et al. [43], they are given by (21).

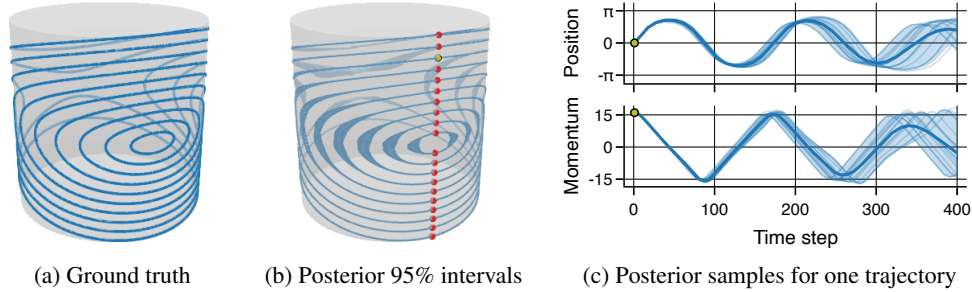

| (a) Ground truth | (b) Posterior 95% intervals | (c) Posterior samples for one trajectory |

Figure 4: Visualization of the dynamical system's learned phase diagram. Middle: we simulate 40 trajectories starting at the red dots, integrate the learned Hamilton's equations forward and backward in time until they approximately intersect other trajectories, and plot 95% intervals in phase space. Right: we simulate the trajectory beginning from the yellow dot, and plot mean and 95% intervals.

## 5 Illustrated Examples

Here we showcase two examples to illustrate the theory: dynamical system prediction and sample path visualization. We focus on simplified settings to present ideas in an easy-to-understand manner.

### 5.1 Dynamical system prediction

We illustrate how Riemannian squared exponential GPs can be used for predicting dynamical systems while respecting the underlying geometry of the configuration space the system is defined on. This is an important task in robotics, where GPs are often trained within a model-based reinforcement learning framework [10, 11]. Here, we consider a purely supervised setup, mimicking the model learning inner loop of said framework.

For a prototype physical system, consider an ideal pendulum, whose configuration space is the circle $\mathbb{S}^1$, and whose phase space is the cotangent bundle $T^*\mathbb{S}^1$, which is isometric to the cylinder $\mathbb{S}^1 \times \mathbb{R}$ equipped with the product metric. The equations of motion are given by Hamilton's equations, which are parameterized by the Hamiltonian $H : T^*\mathbb{S}^1 \to \mathbb{R}$. To learn the equations of motion from observed data, we place a GP prior on the Hamiltonian, with covariance given by a squared exponential kernel on the cylinder, defined as a product kernel of squared exponential kernels on the circle and real line. Following Hensman et al. [21], training proceeds using mini-batch stochastic variational inference with automatic relevance determination. The full setup is given in Appendix A.

To generate trajectories from the learned equations of motion, following Wilson et al. [43], we approximate the prior GP using Fourier features, and employ (2) to transform prior sample paths into posterior sample paths. We then generate trajectories by solving the learned Hamilton's equations numerically for each sample, which is straightforward because the approximate posterior is a basis function approximation and therefore easily differentiated in the ordinary deterministic manner. Results can be seen in Figure 4. From these, we see that our GP learns the correct qualitative behavior of the equations of motion, mirroring the results of Deisenroth and Rasmussen [11].

### 5.2 Sample path visualization

To understand how complicated geometry affects posterior uncertainty estimates and illustrate the techniques on a general Riemannian manifold, we consider a posterior sample path visualization task. We take $M$ to be the *dragon* manifold from the Stanford 3D scanning repository, modified slightly to remove components not connected to the outer surface. We represent the manifold using a 202490-triangle mesh and obtain 500 Laplace–Beltrami eigenpairs numerically using the *Firedrake* package [30].

For training data, we introduce a ground truth function by fixing a distinguished point at the end of the dragon's snout, and compute the sine of the geodesic distance from that point. We then observe this function at 52 points on the manifold chosen from the mesh's nodes, and train a Matérn GP regression model with smoothness $\nu = 3/2$ by maximizing the marginal likelihood with respect to

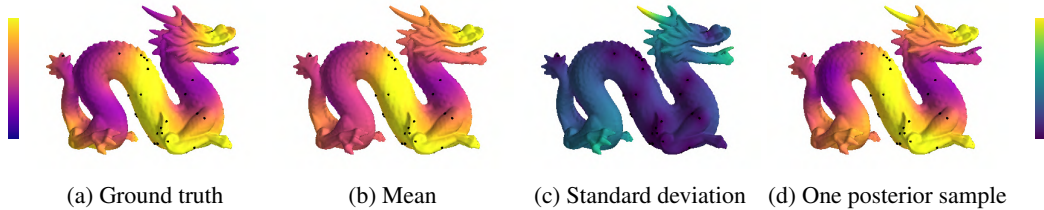

| (a) Ground truth | (b) Mean | (c) Standard deviation | (d) One posterior sample |

Figure 5: Visualization of a Matérn Gaussian process posterior on the dragon. We plot the true function values, posterior mean, marginal posterior variance, and one posterior sample evaluated on the entire mesh. Here, black dots denote training locations, and color represents value of the corresponding functions. Additional posterior samples can be seen in Appendix A.

the remaining kernel hyperparameters. By using the path-wise sampling expression (2), we obtain posterior samples defined on the entire mesh.

Results can be seen in Figure 5. Here, we see that posterior mean and uncertainty estimates match the manifold's shape seamlessly, decaying roughly in proportion with the geodesic distance in most regions. In particular, we see that the two sides of the dragon's snout have very different uncertainty values, despite close Euclidean proximity. This mimics the well-known *swiss roll* example of manifold learning [24, Section 6.1.1], and highlights the value of using a model which incorporates geometry.

## 6  Conclusion

In this work, we developed techniques for computing the kernel, spectral measure, and Fourier feature approximation of Matérn and squared exponential Gaussian processes on compact Riemannian manifolds, thereby constructively generalizing standard Gaussian process techniques to this setting. This was done by viewing the Gaussian processes as solutions of stochastic partial differential equations, and expressing the objects of interest in terms of Laplace–Beltrami eigenvalues and eigenfunctions. The theory was demonstrated on a set of simple examples: learning the equations of motion of an ideal pendulum, and sample path visualization for a Gaussian process defined on a dragon. This illustrates the theory in settings both where Laplace–Beltrami eigenfunctions have a known analytic form, and where they need to be calculated numerically using a differential equation or graphics processing framework. Our work removes limitations of previous approaches, allowing Matérn and squared exponential Gaussian processes to be deployed in mini-batch, online, and non-conjugate settings using variational inference. We hope these contributions enable practitioners in robotics and other physical sciences to more easily incorporate geometry into their models.

## Broader Impact

This is a purely theoretical paper. We develop technical tools that make Matérn Gaussian processes easier to work with in the Riemannian setting. This enables practitioners who are not experts in stochastic partial differential equations to model data that lives on spaces such as spheres and tori.

We envision the impact of this work to be concentrated in the physical sciences, where spaces of this type occur naturally. Since the state spaces of most robotic arms are Riemannian manifolds, we expect these ideas to improve performance of model-based reinforcement learning by making it easier to incorporate geometric prior information into models.

Since climate science is concerned with studying the globe, we also expect that our ideas can be used to model environmental phenomena, such as sea surface temperatures. By employing Gaussian processes for data assimilation and building them into larger frameworks, this could facilitate more accurate climate models compared to current methods.

These impacts carry forward to potential generalizations of our work. We encourage practitioners to consider impacts on their respective disciplines that arise from incorporating geometry into models.

## Acknowledgments and Disclosure of Funding

VB was supported by the St. Petersburg Department of Steklov Mathematical Institute of Russian Academy of Sciences and by the Ministry of Science and Higher Education of the Russian Federation, agreement № 075-15-2019-1620. PM was supported by the Ministry of Science and Higher Education of the Russian Federation, agreement № 075-15-2019-1619. VB and PM were supported by "Native towns", a social investment program of PJSC Gazprom Neft, and by the Department of Mathematics and Computer Science of St. Petersburg State University. AT was supported by the Department of Mathematics at Imperial College London.

## Footnotes

*Equal contribution. Correspondence to: VIACHESLAV.BOROVITSKIY@GMAIL.COM, A.TERENIN17@IMPERIAL.AC.UK, and PMOSTOWSKY@GMAIL.COM.

[2]Note that $\mathbb{T}^2 = \mathbb{S}^1 \times \mathbb{S}^1$ is diffeomorphic but *not* isometric to the usual donut-shaped torus whose metric is induced by embedding in $\mathbb{R}^3$. This is important, because it is the Riemannian metric structure that gives rise to the Laplace–Beltrami operator and hence to the generalized Matérn and squared exponential kernels. Diffeomorphisms do not necessarily preserve metric structure, so they may not preserve kernels.

[3]The marginal variance $k_{(\cdot)}(x, x)$ can depend on $x$, thus we normalize the kernel by the average variance.

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
