[Supplementary Material]

# A  Additional experimental details

**Sample path visualization**

Here, we further explore the example of a Gaussian process on the dragon manifold. Figure 6 presents nine additional samples from Gaussian process posterior. Note that the we change the color palette in order to cover samples' value range. We repeat the first row of Figure 6 with the new color palette.

To define a Matérn kernel on the dragon manifold, we employ a re-parametrized version of (18), which is

$$k_\nu(x, x') = \sigma^2 \sum_{n=0}^{\infty} \left( \frac{1}{\kappa^2} + \lambda_n \right)^{-\nu - \frac{d}{2}} f_n(x) f_n(x'). \tag{23}$$

Here, we need to compute eigenvalues and eigenfunctions of the Laplace–Beltrami operator. We do so using the *Galerkin finite element method* (FEM), and approximate the manifold as a triangular mesh with $K = 100179$ vertices. This involves solving a Helmholtz equation, which is a far easier problem than solving the SPDE 6, since, among other reasons, the equation is a standard deterministic second-order linear PDE that only needs to be solved once, rather than once-per-sample. Each vertex $v_k$, $k = 1, \ldots, K$ is associated with a piecewise-linear basis function $\phi_k$, such that $\phi_k(v_l) = \delta_{kl}$ where $\delta$ is the Kronecker delta. This leads to a *discrete Laplace-Beltrami operator* $\Delta$, as a discretization of the Laplace-Beltrami $\Delta_g$ on the manifold. Finally, the eigenproblem is stated as follows: find $\lambda_n, f_n$, such that

$$\langle \Delta f_n, \phi_k \rangle = \lambda_n \langle f_n, \phi_k \rangle \qquad\qquad k = 1, \ldots, K \tag{24}$$

where we regard the functions $f_n, \phi_k$ as $K$-dimensional vectors: $(f_n)_k = f_n(v_k)$. Since the resulting eigenproblem (24) is finite-dimensional, it can be solved using standard numerical approaches. Note that the discrete Laplacian cannot have more than $K$ eigenvalues. In our experiments, we use the Firedrake software package [30], which provides a high-level domain-specific language for computing discrete Laplacians and related tasks. We use Arnoldi method with shift-invert spectral transform to compute the first $N = 500$ (smallest magnitude) eigenvalues and corresponding eigenfunctions on the dragon mesh, using numerical routines from the PETSc package, which Firedrake calls. This allows us to approximate the formula (18) with the first $N$ components of the sum, given by

$$k_\nu(x, x') \approx \sigma^2 \sum_{n=0}^{N} \left( \frac{1}{\kappa^2} + \lambda_n \right)^{-\nu - \frac{d}{2}} f_n(x) f_n(x'). \tag{25}$$

The formula also leads to a Fourier approximation of the prior:

$$f(x) \approx \sigma \sum_{n=0}^{N} w_n \left( \frac{1}{\kappa^2} + \lambda_n \right)^{-\frac{\nu}{2} - \frac{d}{4}} f_n(x) \qquad\qquad w_n \sim \mathrm{N}(0, 1). \tag{26}$$

Once these expressions are obtained, standard GP training techniques are utilized to compute the posterior distribution using path-wise sampling, given by equation (2). In the experiments we set the smoothness parameter $\nu$ to be $3/2$, and Gaussian noise variance to be $10^{-15}$. We obtain $\sigma^2$ and $\kappa$ by gradient descent optimization of the marginal likelihood.

(a) Ground truth          (b) Mean          (c) Standard deviation

Figure 6: Visualization of a Matérn Gaussian process posterior on the dragon. We plot the true function values, posterior mean, marginal posterior standard deviation, and nine random function draws from the posterior. Here, black dots denote training locations, and color represents value of the corresponding functions. The color palette is changed slightly compared to Figure 5 in order to represent the range of the samples more effectively.

**Dynamical system prediction**

Here we describe the setup in the dynamical systems predictions experiment in more detail. Our system is an ideal pendulum, parameterized by angle and angular momentum which lie on the cylinder $\mathbb{S}^1 \times \mathbb{R}^1$, and which we denote by $(\theta, p_\theta)$. The true equations of motion are given by Hamilton's equations

$$\dot{\theta} = \frac{\partial H}{\partial p_\theta} \qquad\qquad \dot{p} = -\frac{\partial H}{\partial \theta} \qquad\qquad (27)$$

with

$$H(\theta, p_\theta) = \frac{p_\theta^2}{2ml^2} + mgl(1 - \cos(\theta)) \qquad\qquad (28)$$

where $(m, g, l)$ are the mass, gravitational constant, and length of the pendulum. We set $m = 1$, $g = 9.8$, $l = 2$.

Training data is obtained as follows. We do not observe the Hamiltonian: instead, we observe its partial derivative pairs $(\frac{\partial H}{\partial \theta}, \frac{\partial H}{\partial p_\theta})$. In a reinforcement learning setting, following Deisenroth and Rasmussen [11], these can be obtained by backward integration of observed trajectories. In our simplified setting, we generate training data by computing said partial derivatives at random locations, sampled uniformly on the rectangle $(0, 2\pi) \times (-20, 20)$, generating 1000 total training points.

To obtain the model, we first place a Gaussian process prior directly on the Hamiltonian. To ensure that (a) $\theta$ is supported on $[0, 2\pi)$, rather than $[0, 1)$, and (b) a unified random Fourier feature expansion for the prior is possible for both the $\theta$ and $p_\theta$ components, we use a re-parameterized form for the kernel and spectral measure. These are given by

$$k_\theta(\theta, \theta') = \sum_{n \in 2\pi \mathbb{Z}^d} \exp\left(-\left\|\frac{\theta - \theta' + n}{2\pi\sqrt{2}\kappa}\right\|^2\right) = \sum_{n \in 2\pi \mathbb{Z}^d} \exp\left(-\left\|\frac{\theta - \theta' + n}{\kappa_\theta}\right\|^2\right) \qquad (29)$$

$$\rho_\theta(n) = \sqrt{2\pi}2^{-3/2}\pi^{-1}\kappa_\theta \exp(-2\pi^2 2^{-3}\pi^{-2}\kappa_\theta^2 n^2) = 2^{-1}\pi^{-1/2}\kappa_\theta \exp(-2^{-2}\kappa_\theta^2 n^2) \qquad (30)$$

which from the kernel and spectral measure introduced in the manuscript by defining $\theta = 2\pi x$, $\theta' = 2\pi x'$, and $\kappa_\theta = 2^{3/2}\pi\kappa$ so that $\kappa = 2^{-3/2}\pi^{-1}\kappa_\theta$. For the $p_\theta$ component, we use the re-normalized squared exponential kernel

$$k_{p_\theta}(p_\theta, p'_\theta) = \exp\left(-\left\|\frac{p_\theta - p'_\theta}{\kappa_{p_\theta}}\right\|^2\right) \qquad\qquad (31)$$

and denote the corresponding spectral measure by $\rho_{p_\theta}$. The full kernel on $(\theta, p_\theta)$ is given by

$$k\big((\theta, p_\theta), (\theta', p'_\theta)\big) = \sigma^2 k_\theta(\theta, \theta') k_{p_\theta}(p_\theta, p'_\theta). \qquad\qquad (32)$$

Similarly, denote the full spectral measure over $\mathbb{Z} \times \mathbb{R}$ by $\rho$. Sampling from the posterior is performed by sampling from the prior using a random Fourier feature approximation and transforming the resulting draws into posterior draws using (2).

Unfortunately, since the spectral measure for this kernel is the product of a discrete measure for the $\theta$ component, and absolutely continuous measure for the $p_\theta$ component, the resulting optimization objective is not (automatically) differentiable with respect to $\kappa_{p_\theta}$. To enable use of automatic relevance determination, we develop an importance-sampling-based *reparametrization trick* by employing the generalized random Fourier feature expansion

$$f(\theta, p_\theta) \approx \sigma\sqrt{\frac{2}{\ell}}\sum_{j=1}^{\ell} \gamma_j w_j \cos\left(\left\langle \boldsymbol{\omega}_j, \frac{(\theta, p_\theta)}{\boldsymbol{\lambda}}\right\rangle + \beta_j\right) \qquad (33)$$

where division by $\boldsymbol{\lambda} = (1, \kappa_{p_\theta})$ is performed element-wise, and

$$\boldsymbol{\omega}_j \sim \widehat{\rho} \qquad\qquad \beta_j \sim \mathrm{U}(0, 2\pi) \qquad\qquad w_j \sim \mathrm{N}(0, 1) \qquad (34)$$

where $\widehat{\rho}$ is the *standard spectral measure*, which is equal to $\rho$ except with $\kappa_\theta$ and $\kappa_{p_\theta}$ fixed to reference values, in our case $\kappa_\theta = \kappa_{p_\theta} = 1$. The importance weights $\gamma_j$ are given by

$$\gamma_j = \sqrt{\frac{\rho_\theta(\omega_{j\theta})/C_\theta}{\widehat{\rho}_\theta(\omega_{j\theta})/\widehat{C}_\theta}} = \sqrt{\frac{\rho_\theta(\omega_{j\theta})\widehat{C}_\theta}{\widehat{\rho}_\theta(\omega_{j\theta})C_\theta}} \qquad\qquad (35)$$

where $\omega_{j\theta}$ is the $\theta$-component of $\boldsymbol{\omega}_j$ and $C_\theta = \sum_{n\in\mathbb{Z}} \rho_\theta(n)$ and $\widehat{C}_\theta = \sum_{n\in\mathbb{Z}} \widehat{\rho}_\theta(n)$ are their respective normalizing constants. Using this more general random Fourier feature approximation, the training objective becomes differentiable with respect to $\kappa_{p_\theta}$.

Since we do not observe the Hamiltonian, but rather its partial derivatives $(\frac{\partial H}{\partial \theta}, \frac{\partial H}{\partial p_\theta})$, as our full model we employ the *gradient* of the Gaussian process developed above, which yields a vector-valued Gaussian process. The kernel of said process is obtained by differentiating the kernels above.

To complete the model, we now introduce the inducing point approximation. We use a total of 35 vector-valued inducing points, which are initialized on an evenly-spaced grid over the domain of the training data. For the prior approximation, we use a total of 128 random Fourier features.

Following Titsias [41] and Hensman et al. [21], training proceeds by minimizing Kullback-Leibler divergence between the inducing point GP and the true posterior GP. We optimize the inducing points, inducing covariance, and all model hyperparameters. For the loss, in addition to the KL divergence, we include $\ell^2$ regularization terms corresponding to log-normal hyperpriors for the hyperparameters. For the kernel, these are given as $\sigma^2 \sim \mathrm{LN}(0,1)$, $\kappa_\theta \sim \mathrm{LN}(0,1)$ and $\ln \kappa_{p_\theta} \sim \mathrm{LN}(1.5,1)$. For the GP, the error variance hyperprior $\tau^2 \sim \mathrm{LN}(10^{-12},1)$. All parameters are initialized at their hyperprior's mean. The jitter term is set to $\varsigma = 10^{-5}$.

Optimization is performed by the ADAM algorithm, with learning rate set to $\eta = 0.01$ and default values for the other hyperparameters. We use a mini-batch size of 128, and train until convergence.

To generate trajectories of the dynamical system under the learned Hamiltonian, following Wilson et al. [43], we use (2) to draw a set of basis coefficients from the posterior distribution, and form a basis function approximation of our posterior GP. We plug this function back into Hamilton's equations, and solve them numerically by employing a Störmer-Verlet integrator. The step size is tuned for each initial condition to ensure all trajectories in Figure 4 cross each other on the rear side of the cylinder at approximately the same time when using the true Hamiltonian after 50 time steps, and range from 0.02 to 0.031. These step sizes are then used to produce error bars for the learned Hamiltonian.

To generate the error bars on the cylinder Figure 4, we first compute the mean trajectory under the GP model for each time step. Then, for each time step, we project the trajectories onto the tangent plane on the cylinder located at the mean, using the cross product identity. In this tangent plane, we then project the trajectory points onto a line perpendicular to the tangent vector pointing in the direction of the mean trajectory obtained by backwards integration. We calculate 95% intervals over this line, and plot them projected back from the tangent plane onto the surface of the cylinder. The error bars for the positions and momenta of the distinguished trajectory on the right-hand-side of Figure 4, which are not plotted on the surface of the cylinder, are obtained by re-parameterizing $\vartheta = ((\theta + \pi)$ mod $2\pi) - \pi$ to ensure $\vartheta \in [-\pi, \pi)$, and calculating 2.5% and 97.5% quantiles in the standard way.

## B  Additional examples and expressions

**Circle**

Here we discuss closed-form expressions for Matérn and squared exponential kernels on circle $\mathbb{S}^1 = \mathbb{T}$. These kernels are given in (9) and (10) respectively, with $d = 1$ in our setting. Applying the generalized Poisson summation formula [39, Chapter VIII] to these expressions gives

$$k_\nu(x,x') = \sum_{n\in\mathbb{Z}} \frac{S_\nu(n)}{C'_\nu} e^{2\pi in(x-x')}, \qquad k_\infty(x,x') = \sum_{n\in\mathbb{Z}} \frac{S_\infty(n)}{C'_\infty} e^{2\pi in(x-x')}, \qquad (36)$$

where $S_\nu$ and $S_\infty$ are precisely the spectral densities of the standard Matérn and squared exponential kernels over $\mathbb{R}$. The specific formulas for $S_\nu, S_\infty$ are given in Rasmussen and Williams [29, Section 4.2.1]:

$$S_\nu(\xi) = \sigma^2 \frac{2\pi^{\frac{1}{2}}\Gamma(\nu + \frac{1}{2})(2\nu)^\nu}{\Gamma(\nu)\kappa^{2\nu}} \left( \frac{2\nu}{\kappa^2} + 4\pi^2\xi^2 \right)^{-\left(\nu + \frac{1}{2}\right)}, \qquad (37)$$

$$S_\infty(\xi) = \sigma^2(2\pi\kappa^2)^{1/2} e^{-2\pi^2\kappa^2\xi^2}, \qquad (38)$$

where the cumbersome constants ensure that the original GP over $\mathbb{R}$ has variance equal to $\sigma^2$. Periodic summation does not preserve variance, thus requiring additional constants $C'_{(\cdot)}$ to recover variance $\sigma^2$. This makes the original constants redundant, so we instead consider

$$\tilde{k}_\nu(x, x') = \sum_{n\in\mathbb{Z}} \left(\frac{2\nu}{\kappa^2} + 4\pi^2 n^2\right)^{-\left(\nu+\frac{1}{2}\right)} e^{2\pi i n \cdot (x-x')}, \tag{39}$$

$$\tilde{k}_\infty(x, x') = \sum_{n\in\mathbb{Z}} e^{-2\pi^2\kappa^2 n^2} e^{2\pi i n \cdot (x-x')}. \tag{40}$$

For $\nu = \infty$ the right-hand side is precisely one of the classical Jacobi theta functions, $\vartheta_3(z, q)$ (see definition in Abramowitz and Stegun [1, equation 16.27.3]), with parameters $z = \pi(x - x')$ and $q = \exp(-2\pi^2\kappa^2)$, giving

$$\tilde{k}_\infty(x, x') = \vartheta_3(\pi(x - x'), \exp(-2\pi^2\kappa^2)). \tag{41}$$

To obtain $k_\infty$ from $\tilde{k}_\infty$ we need to find $C_\infty$ such that $\tilde{k}_\infty(x, x)/C_\infty = \sigma^2$. Obviously, $C_\infty = \tilde{k}_\infty(x, x)/\sigma^2$, where the right hand side does not depend on $x$, so the constant is well-defined. Hence

$$k_\infty(x, x') = \frac{\sigma^2}{\vartheta_3(0, \exp(-2\pi^2\kappa^2))} \vartheta_3(\pi(x - x'), \exp(-2\pi^2\kappa^2)). \tag{42}$$

Returning to the periodic summation of the original normalized kernel, we obtain

$$C'_\infty = C_\infty \sigma^2 (2\pi\kappa^2)^{1/2} = \vartheta_3(0, \exp(-2\pi^2\kappa^2))(2\pi\kappa^2)^{1/2}. \tag{43}$$

Summarizing, we obtain the following.

**Example 7** (Squared exponential kernel on $\mathbb{S}^1$). *The squared exponential kernel, normalized to have variance $\sigma^2$, and the corresponding spectral density, are given by*

$$k_\infty(x, x') = \frac{\sigma^2}{\vartheta_3(0, \exp(-2\pi^2\kappa^2))} \vartheta_3(\pi(x - x'), \exp(-2\pi^2\kappa^2)), \tag{44}$$

$$\rho_\infty(n) = \frac{\sigma^2}{\vartheta_3(0, \exp(-2\pi^2\kappa^2))} \exp(-2\pi^2\kappa^2 n^2), \qquad n \in \mathbb{Z}. \tag{45}$$

Now we turn our attention to kernels $k_\nu$. After an appropriate *mutatis mutandis* applied to the closed-form of the Fourier series

$$\frac{\alpha \sinh(\alpha\pi)}{\pi} \sum_{k=-\infty}^{\infty} \frac{\exp(ik\theta)}{(\alpha^2 + k^2)^n} \tag{46}$$

provided in the supplementary material of Guinness and Fuentes [19] we get the following.

**Example 8** (Matérn kernel on $\mathbb{S}$ for half-integer $\nu$). *Let $\nu = 1/2 + s$, $s \in \mathbb{N}$. The Matérn kernel, normalized to have variance $\sigma^2$, and the corresponding spectral density, are given by*

$$k_\nu(x, x') = \frac{\sigma^2}{C_\nu} \sum_{k=0}^{s} a_{s,k} \left(\sqrt{2\nu} \cdot \frac{|x - x'| - 1/2}{\kappa}\right)^k \text{hyp}^k\left(\sqrt{2\nu} \cdot \frac{|x - x'| - 1/2}{\kappa}\right) \tag{47}$$

$$\rho_\nu(n) = \frac{2\sigma^2\sqrt{2\nu}\sinh\left(\frac{\sqrt{2\nu}}{2\kappa}\right)}{C_\nu(2\pi)^{1-2\nu}\kappa} \left(\frac{2\nu}{\kappa^2} + 4\pi^2 n^2\right)^{-\nu-1/2}, \qquad n \in \mathbb{Z} \tag{48}$$

*where various components of the expression are defined as follows.*

1. *$\text{hyp}^k(\cdot)$ is defined as $\cosh(\cdot)$ when $k$ is odd and $\sinh(\cdot)$ when $k$ is even.*

2. *$C_\nu$ is chosen so that $k_\nu(x, x) = \sigma^2$.*

3. *$a_{s,k}$ are constants defined as follows, following a modification of the derivation given by Guinness and Fuentes [19].*

(a) First, for the special case $k = s$, define

$$a_{s,s} = \left(\left(-\frac{\nu}{\pi^2\kappa^2}\right)^s (s)!\right)^{-1}. \tag{49}$$

(b) Next, define the constants $h_{rk}$ as

$$h_{rk} = \sum_{j=0}^{2r+1} \binom{2r+1}{j} (k)_j \left(\frac{\sqrt{2\nu}}{2\kappa}\right)^{k-j} \cdot \mathrm{hyp}^{k-j+1}\left(\frac{\sqrt{2\nu}}{2\kappa}\right) \tag{50}$$

for $r = 0, \ldots, s-1$ and $k = 0, \ldots, s$, where $(k)_j$ is the falling factorial

$$(k)_j = \begin{cases} 1, & \text{when } j = 0, \\ 0, & \text{when } j > k, \\ k(j-1)\ldots(k-j+1), & \text{otherwise.} \end{cases} \tag{51}$$

(c) Finally, define the matrix

$$\mathbf{H}_s = (h_{rk})_{r=0,\ldots,s-1}^{k=0,\ldots,s-1}, \tag{52}$$

and a vector $\mathbf{h}_s = [h_{0,s}, \ldots h_{s-1,s}]^\top$. Then the remaining constants $a_{s,k}$ for $k \neq s$ are given as

$$[a_{s,0}, \ldots, a_{s,s-1}]^\top = -a_{s,s}\mathbf{H}_s^{-1}\mathbf{h}_s. \tag{53}$$

For $\nu = 1/2$ the above formulae reduce to

$$k_{1/2}(x, x') = \frac{\sigma^2}{\cosh\left(\frac{1}{2\kappa}\right)} \cosh\left(\frac{|x - x'| - 1/2}{\kappa}\right), \tag{54}$$

$$\rho_{1/2}(n) = \frac{\sigma^2 2\sinh\left(\frac{1}{2\kappa}\right)}{\kappa \cosh\left(\frac{1}{2\kappa}\right)} \left(\frac{1}{\kappa^2} + 4\pi^2 n^2\right)^{-1}. \tag{55}$$

For $\nu = 3/2$,

$$k_{3/2}(x, x') = \frac{\sigma^2}{C_{3/2}} \left(\frac{\pi^2\kappa}{3}\left(2\kappa + \sqrt{3}\coth\left(\frac{\sqrt{3}}{2\kappa}\right)\right)\cosh(u) - \frac{2\pi^2\kappa^2}{3}u\sinh(u)\right), \tag{56}$$

$$\rho_{3/2}(n) = \frac{\sigma^2}{C_{3/2}} \frac{2\sqrt{3}\sinh\left(\frac{\sqrt{3}}{2\kappa}\right)}{(2\pi)^{-2}\kappa} \left(\frac{3}{\kappa^2} + 4\pi^2 n^2\right)^{-2}, \tag{57}$$

where $u = \sqrt{3}\frac{|x-x'| - 1/2}{\kappa}$.

For $\nu = 5/2$, they reduce to

$$k_{5/2}(x, x') = \frac{\sigma^2}{C_{5/2}} \left(a_{2,0}\cosh(u) + a_{2,1}u\sinh(u) + a_{2,2}u^2\cosh(u)\right), \tag{58}$$

$$\rho_{5/2}(n) = \frac{\sigma^2}{C_{5/2}} \frac{2\sqrt{5}\sinh\left(\frac{\sqrt{5}}{2\kappa}\right)}{(2\pi)^{-4}\kappa} \left(\frac{5}{\kappa^2} + 4\pi^2 n^2\right)^{-3}, \tag{59}$$

where $u = \sqrt{5}\frac{|x-x'| - 1/2}{\kappa}$ and

$$a_{2,0} = -\frac{\pi^4\kappa^2}{50}\left(-5 + 12\kappa^2 + 6\sqrt{5}\kappa\coth\left(\frac{\sqrt{5}}{2\kappa}\right) + 10\coth\left(\frac{\sqrt{5}}{2\kappa}\right)^2\right), \tag{60}$$

$$a_{2,1} = \frac{2\pi^4\kappa^3}{25}\left(3\kappa + \sqrt{5}\coth\left(\frac{\sqrt{5}}{2\kappa}\right)\right), \qquad a_{2,2} = -\frac{2\pi^4\kappa^4}{25}. \tag{61}$$

Finally, we discuss Fourier feature approximations. The main ingredient of these approximations is the formula

$$k_{(\cdot)}(x, x') = \sum_{n \in \mathbb{Z}} \rho_{(\cdot)}(n) e^{2\pi i n(x-x')} = \rho_{(\cdot)}(0) + 2 \sum_{n \in \mathbb{N}} \rho_{(\cdot)}(n) \cos(2\pi n(x - x')). \quad (62)$$

The sum on the right hand side can be approximated either deterministically by truncating the series, or randomly with Monte Carlo techniques. This corresponds respectively to the following two approximations of the process

$$f_{(\cdot)}^D(x) = \sum_{n=-N}^{N} \sqrt{\rho_{(\cdot)}(n)} \big(w_{n,1} \cos(2\pi n x) + w_{n,2} \sin(2\pi n x)\big), \qquad w_{n,j} \sim \mathrm{N}(0,1), \quad (63)$$

and

$$f_{(\cdot)}^R(x) = \frac{\sigma}{\sqrt{N}} \sum_{k=0}^{N-1} \big(w_{n,1} \cos(2\pi n_k x) + w_{n,2} \sin(2\pi n_k x)\big), \qquad n_k \sim \frac{\rho_{(\cdot)}(n)}{\sigma^2}, \quad (64)$$

where $w_{n,j}$ is defined identically. Note that the kernel discussed here is defined via periodic summation defined in Section 3.

## Sphere

The presentation here is based on De Vito et al. [9, Section 7.3]. Assume $d > 1$ and take $M = \mathbb{S}^d$, where $\mathbb{S}^d$ is $d$-dimensional sphere $\mathbb{S}^d \subseteq \mathbb{R}^{d+1}$. The 1-dimensional case discussed in the previous section can be handled similarly but requires some additional care.

The eigenvalues of $\Delta_{\mathbb{S}^d}$ are $\lambda_n = n(n + d - 1), n \in \mathbb{Z}_+$. The eigenspace $\mathcal{H}_n$ corresponding to $\lambda_n$ has dimension $d_n = (2n + d - 1)\frac{\Gamma(n+d-1)}{\Gamma(d)\Gamma(n+1)}$ and consists of spherical harmonics of degree $n$. The addition formula for spherical harmonics yields that for any orthonormal basis $f_{n,k}$ of eigenspace $\mathcal{H}_n$

$$\sum_{k=1}^{d_n} f_{n,k}(x) f_{n,k}(x') = c_{n,d} \mathcal{C}_n^{(d-1)/2}(\cos(d_M(x, x'))) \quad (65)$$

where $\mathcal{C}_n^{(d-1)/2}$ are Gegenbauer polynomials and the constant $c_{n,d}$ is defined by

$$c_{n,d} = \frac{d_n \Gamma((d+1)/2)}{2\pi^{(d+1)/2} \mathcal{C}_n^{(d-1)/2}(1)}. \quad (66)$$

From this we deduce that the formula for Matérn kernel on $\mathbb{S}^d$ is given by

$$k_\nu(x, x') = \frac{\sigma^2}{C_\nu} \sum_{n=0}^{\infty} \left(\frac{2\nu}{\kappa^2} + \lambda_n\right)^{-(\nu+\frac{d}{2})} \left(\sum_{k=1}^{d_n} f_{n,k}(x) f_{n,k}(x')\right) \quad (67)$$

$$= \frac{\sigma^2}{C_\nu} \sum_{n=0}^{\infty} \left(\frac{2\nu}{\kappa^2} + n(n + d - 1)\right)^{-(\nu+\frac{d}{2})} c_{n,d} \mathcal{C}_n^{(d-1)/2}(\cos(d_M(x, x'))). \quad (68)$$

Analogously for squared exponential kernel on $\mathbb{S}^d$, we obtain

$$k_\infty(x, x') = \frac{\sigma^2}{C_\infty} \sum_{n=0}^{\infty} e^{-\frac{\kappa^2}{2}\lambda_n} \left(\sum_{k=1}^{d_n} f_{n,k}(x) f_{n,k}(x')\right) \quad (69)$$

$$= \frac{\sigma^2}{C_\infty} \sum_{n=0}^{\infty} e^{-\frac{\kappa^2}{2}n(n+d-1)} c_{n,d} \mathcal{C}_n^{(d-1)/2}(\cos(d_M(x, x'))). \quad (70)$$

Summarizing the above, for the sphere $\mathbb{S}^d$ we obtain the following.

**Example 9** (Matérn and squared exponential kernels on $\mathbb{S}^d$)**.** *The Matérn and squared exponential kernels and the corresponding spectral densities are given as follows*

$$k_\nu(x, x') = \frac{\sigma^2}{C_\nu} \sum_{n=0}^\infty \left( \frac{2\nu}{\kappa^2} + n(n+d-1) \right)^{-\left(\nu + \frac{d}{2}\right)} c_{n,d} \mathcal{C}_n^{(d-1)/2}(\cos(d_M(x, x'))), \qquad (71)$$

$$k_\infty(x, x') = \frac{\sigma^2}{C_\infty} \sum_{n=0}^\infty e^{-\frac{\kappa^2}{2} n(n+d-1)} c_{n,d} \mathcal{C}_n^{(d-1)/2}(\cos(d_M(x, x'))), \qquad (72)$$

$$\rho_\nu(n) = \frac{\sigma^2}{C_\nu} \left( \frac{2\nu}{\kappa^2} + n(n+d-1) \right)^{-\left(\nu + \frac{d}{2}\right)}, \qquad (73)$$

$$\rho_\infty(n) = \frac{\sigma^2}{C_\infty} e^{-\frac{\kappa^2}{2} n(n+d-1)}, \qquad (74)$$

*where $d_M(x, x')$ is the geodesic distance between $x, x' \in \mathbb{S}^d$, $\mathcal{C}_n^{(d-1)/2}$ are Gegenbauer polynomials and*

$$c_{n,d} = \frac{d_n \Gamma((d+1)/2)}{2\pi^{(d+1)/2} \mathcal{C}_n^{(d-1)/2}(1)} \qquad with \qquad d_n = (2n+d-1) \frac{\Gamma(n+d-1)}{\Gamma(d)\Gamma(n+1)}. \qquad (75)$$

*Note that for every $n \in \mathbb{Z}_+$ there are $d_n$ Laplace–Beltrami eigenfunctions. Thus, in the following Fourier feature approximation, we cannot apply the combinatorial simplification that yields the Gegenbauer polynomials, and instead work with spherical harmonics directly. The generalized Fourier feature approximations, both deterministic and random, are given by*

$$f_{(\cdot)}^D(x) = \sum_{n=0}^{N-1} \sqrt{\rho_{(\cdot)}(n)} \sum_{j=1}^{d_n} w_{n,j} f_{n,j}(x), \qquad w_{n,j} \sim \mathrm{N}(0,1), \qquad (76)$$

*and*

$$f_{(\cdot)}^R(x) = \frac{\sigma}{\sqrt{N}} \sum_{k=0}^{N-1} \sum_{j=1}^{d_{n_k}} w_{n_k,j} f_{n_k,j}(x), \qquad n_k \sim \frac{\rho_{(\cdot)}(n)}{\sigma^2}, \qquad w_{n_k,j} \sim \mathrm{N}(0,1), \qquad (77)$$

*where $f_{n,k}$ are the actual spherical harmonics forming the orthonormal basis of eigenspace $\mathcal{H}_n$.*

No closed form expressions for $k_\nu$ and $k_\infty$ are known to the authors. Nevertheless, approximating the series defining $k_\nu$ and $k_\infty$ by truncation gives a practical approach with reasonable error control. Note that the larger $\nu$ is, the faster these series converge, and the more accurate the resulting approximations are.

## C  Proof of Proposition 2

**Proposition 2.** *The Matérn (squared exponential) kernel k in* (9) *(resp.* (10)*) is the covariance kernel of the Matérn (resp. squared exponential) Gaussian process in the sense of Whittle [42].*

*Proof.* Following Section 4, the Matérn and square exponential kernels on a compact Riemannian manifold in the sense of Whittle [42] are given by (18) and (19). For the sake of this proof we denote these kernels by $k_{(\cdot)}^{(w)}$ and the kernels defined by periodic summation (equations (9), (10)) by $k_{(\cdot)}^{(p)}$. We prove here that $k_{(\cdot)}^{(p)}$ are equal to $k_{(\cdot)}^{(w)}$.

To make equations (18) and (19) explicit for $\mathbb{T}^d = \mathbb{R}^d/\mathbb{Z}^d$, we need to compute the eigenfunctions and eigenvalues of Laplace–Beltrami operator $\Delta_g$ on $\mathbb{T}^d$. This is not difficult, since $\mathbb{T}^d$ is equipped with the quotient metric, which is flat. In particular, this amounts to considering the eigenfunctions of Euclidean Laplacian, which are sines and cosines (complex exponentials), and leaving only those which are 1-periodic. The procedure is described in detail in Gordon [16], and yields the following. For $\tau \in \mathbb{Z}_+^d, \tau \neq 0$ the pair of functions $f_{\tau,1}(x) = \sqrt{2}\cos(2\pi\tau \cdot x)$ and $f_{\tau,2}(x) = \sqrt{2}\sin(2\pi\tau \cdot x)$ are eigenfunctions of the Laplace–Beltrami operator, corresponding to the eigenvalue

$\lambda_\tau = 4\pi^2|\tau|^2$. It is important to note that $\tau$ and $-\tau$ correspond to the same (up to a sign change) pair of eigenfunctions. Together with the function $f_0(x) = 1$ corresponding to the eigenvalue $\lambda_0 = 0$, they form the orthonormal basis of $L^2(\mathbb{T}^d)$. To unify notation, we write $f_{0,1}(x) = 1$ and $f_{0,2}(x) = 0$.

Since the series defining $k_{(\nu)}^{(w)}$ is unconditionally convergent [9], we obtain

$$k_\nu^{(w)}(x,x') = \frac{\sigma^2}{C_\nu} \sum_{\tau \in \mathcal{I}} \left( \frac{2\nu}{\kappa^2} + 4\pi^2|\tau|^2 \right)^{-\nu - \frac{d}{2}} (f_{\tau,1}(x)f_{\tau,1}(x') + f_{\tau,2}(x)f_{\tau,2}(x')), \qquad (78)$$

where $\mathcal{I} \subseteq \mathbb{Z}^d$ is a maximal subset of $\mathbb{Z}^d$ such that $\tau \in \mathcal{I}$ and $\tau \neq 0$ implies $-\tau \notin \mathcal{I}$. This, using identities $\cos(x-y) = \cos(x)\cos(y) + \sin(x)\sin(y)$ and $\cos(x) = (\cos(x) + \cos(-x))/2$, becomes

$$k_\nu^{(w)}(x,x') = \frac{\sigma^2}{C_\nu} \sum_{\tau \in \mathbb{Z}^d} \left( \frac{2\nu}{\kappa^2} + 4\pi^2|\tau|^2 \right)^{-\nu - \frac{d}{2}} \cos(2\pi\tau \cdot (x - x')). \qquad (79)$$

At the same time, the generalized Poisson summation formula gives

$$k_\nu^{(p)}(x,x') = \frac{\sigma^2}{C'_\nu} \sum_{n \in \mathbb{Z}^d} S(n)e^{2\pi in\cdot(x-x')} = \frac{\sigma^2}{C'_\nu} \left( \sum_{\tau \in \mathbb{Z}^d} S(\tau)\cos(2\pi\tau \cdot (x - x')) \right), \qquad (80)$$

where $S$ is the spectral density of Matérn kernel on $\mathbb{R}^d$. This is given by [29, Section 4.2.1]

$$S(\xi) = \frac{2^d\pi^{\frac{d}{2}}\Gamma(\nu + \frac{d}{2})(2\nu)^\nu}{\Gamma(\nu)\kappa^{2\nu}} \left( \frac{2\nu}{\kappa^2} + 4\pi^2|\xi|^2 \right)^{-(\nu+\frac{d}{2})}. \qquad (81)$$

Thus for finite $\nu$ we have

$$k_\nu^{(p)}(x,x') = \frac{C_\nu 2^d\pi^{\frac{d}{2}}\Gamma(\nu + \frac{d}{2})(2\nu)^\nu}{C'_\nu\Gamma(\nu)\kappa^{2\nu}} k_\nu^{(w)}(x,x'). \qquad (82)$$

Recalling that $C_\nu$ and $C'_\nu$ are chosen so that $k_\nu^{(p)}(x,x) = \sigma^2 = k_\nu^{(w)}(x,x)$, we see that $k_\nu^{(p)}(x,x') = k_\nu^{(w)}(x,x')$, which gives the claim.

The argument for squared exponential kernel ($\nu = \infty$) is essentially the same. In this case we have

$$k_\infty^{(w)}(x,x') = \frac{\sigma^2}{C_\infty} \sum_{\tau \in \mathbb{Z}^d} \exp\left( -2\pi^2\kappa^2|\tau|^2 \right) \cos(2\pi\tau \cdot (x - x')), \qquad (83)$$

$$k_\infty^{(p)}(x,x') = \frac{\sigma^2}{C'_\infty} \left( \sum_{\tau \in \mathbb{Z}^d} S(\tau)\cos(2\pi\tau \cdot (x - x')) \right), \qquad (84)$$

but this time with

$$S(\xi) = \sigma^2(2\pi\kappa^2)^{d/2}e^{-2\pi^2\kappa^2|\xi|^2}. \qquad (85)$$

This gives

$$k_\infty^{(p)}(x,x') = \frac{C_\infty(2\pi\kappa^2)^{d/2}}{C'_\infty} k_\infty^{(w)}(x,x'), \qquad (86)$$

which translates into $k_\infty^{(p)}(x,x') = k_\infty^{(w)}(x,x')$ with our specific choice of constants $C_\infty$ and $C'_\infty$, and thus completes the proof. $\qquad\square$

## D Theory: compact Riemannian manifolds without boundary

Here we introduce an appropriate formalism for the stochastic partial differential equations (6) and (7) and prove that their solutions are the reproducing kernels of the Sobolev and diffusion spaces given by De Vito et al. [9].

Let $(M, g)$ be a compact connected Riemannian manifold without boundary,[4] and let $\Delta_g$ be the Laplace–Beltrami operator defined on the space $C^\infty(M)$ of smooth functions on $M$. Let $L^2(M)$ denote the space of (almost everywhere equal equivalence classes of) functions on $M$ which are square integrable with respect to the Riemannian volume measure.

**Theorem 11.** *The operator* $-\Delta_g : C^\infty(M) \to L^2(M)$ *uniquely extends to a self-adjoint unbounded operator from some domain* $D(\Delta_g) \subseteq L^2(M)$ *to* $L^2(M)$, *and this extension, denoted again by* $-\Delta_g$, *is a positive operator.*

*Proof.* Strichartz [40, Theorem 2.4]. □

This allows one to apply the spectral theorem for self-adjoint unbounded operators, which, loosely speaking, diagonalizes such operators and enables us to introduce a functional calculus for them. The general statement of the spectral theorem for unbounded self-adjoint operators can be found in various textbooks—see, for instance, Lang [23, Chapters XIX and XX] or Reed and Simon [31, Chapter VIII]. For our setting, we do not need this general statement, as there is a separate theorem for the special case of the Laplace–Beltrami operator on a compact manifold, commonly referred to as the Sturm-Liouville decomposition.

**Theorem 12** (Sturm–Liouville decomposition). *Let* $(M, g)$ *be a compact Riemannian manifold without boundary. Then there exists an orthonormal basis* $\{f_n\}_{n \in \mathbb{Z}_+}$, *of* $L^2(M)$ *such that* $-\Delta_g f_n = \lambda_n f_n$ *with* $0 = \lambda_0 < \lambda_1 \leq .. \leq \lambda_n$ *and* $\lambda_n \to \infty$ *as* $n \to \infty$. *Moreover,* $-\Delta_g$ *admits the representation*

$$-\Delta_g f = \sum_{n=0}^\infty \lambda_n \langle f, f_n \rangle f_n, \tag{87}$$

*which converges unconditionally in* $L^2(M)$ *for all* $f \in D(\Delta_g)$.

*Proof.* See Chavel [7, page 139] or Canzani [6, Theorem 44]. □

This allows one to define a (possibly unbounded) operator $\Phi(-\Delta_g)$ for any Borel measurable function $\Phi : [0, +\infty) \to \mathbb{R}$ by

$$\Phi(-\Delta_g)f = \sum_{n=0}^\infty \Phi(\lambda_n)\langle f, f_n \rangle f_n \tag{88}$$

with domain given by

$$D(\Phi(-\Delta_g)) = \left\{ f \in L^2(M) \,\middle|\, \sum_{n=0}^\infty |\Phi(\lambda_n)|^2 |\langle f, f_n \rangle|^2 < \infty \right\}. \tag{89}$$

This idea is called the functional calculus for the operator $-\Delta_g$. It allows us to formally define operators from the SPDEs under consideration with

$$\left( \frac{2\nu}{\kappa^2} - \Delta_g \right)^{\frac{\nu}{2}+\frac{d}{4}} f = \sum_{n=0}^\infty \left( \frac{2\nu}{\kappa^2} + \lambda_n \right)^{\frac{\nu}{2}+\frac{d}{4}} \langle f, f_n \rangle f_n, \quad \text{using } \Phi(\lambda) = \left( \frac{2\nu}{\kappa^2} + \lambda \right)^{\frac{\nu}{2}+\frac{d}{4}}, \tag{90}$$

$$e^{-\frac{\kappa^2}{4}\Delta} f = \sum_{n=0}^\infty e^{\frac{\kappa^2 \lambda_n}{4}} \langle f, f_n \rangle f_n, \quad \text{using } \Phi(\lambda) = e^{\frac{\kappa^2 \lambda}{4}}. \tag{91}$$

Denote these operators by $\mathcal{L}$. We now proceed to define an appropriate formalism for the SPDEs

$$\mathcal{L}f = \mathcal{W}. \tag{92}$$

We start by introducing a notion of generalized Gaussian random fields.

**Definition 13** (Definition 3.2.10 of Lototsky and Rozovsky [26]). *A zero-mean generalized Gaussian field* $\mathrm{F}$ *over a Hilbert space* $H$ *is a collection of Gaussian random variables* $\{\mathrm{F}(h)\}_{h \in H}$ *with the properties*

1. $\mathbb{E}(\mathrm{F}(h)) = 0$ *for all* $h \in H$,

2. *There exists a bounded, linear, self-adjoint, non-negative operator* $K$ *on* $H$ *(called the covariance operator of* $\mathrm{F}$) *such that*

$$\mathbb{E}(\mathrm{F}(h)\,\mathrm{F}(g)) = \langle Kh, g \rangle_H \tag{93}$$

*for all* $h, g \in H$.

A zero-mean generalized Gaussian field $\mathcal{W}$ over a Hilbert space $H$ with identity $I : H \to H$ serving as covariance operator is called the *standard Gaussian white noise* over $H$.

Let $\mathcal{W}$ be said white noise over $L^2(M)$. Up to a normalizing constant which ensures that the solution has the right variance, this is equal to the right hand side of equation (92). We do not dwell on this constant until the very end of this section, where it appears naturally as the normalizing constant of the resulting kernel.

It is easy to see that the generalized Gaussian field which we have just defined can be thought of as an operator from $H$ to the space $L^2(\Omega)$ of zero mean random variables with finite variance. From this view, the Gaussian white noise $\mathcal{W}$ is an isometric embedding.

To give more intuition, we explicitly consider how the usual concept of a Gaussian process embeds into this generalization. Let $f \sim \mathrm{GP}(0, k(x, x'))$ be a Gaussian process over a manifold $M$ with covariance function $k(x, x')$. Assume that $k$ is regular enough to consider samples of $f$ as elements of $L^2(M)$. Almost every practically reasonable covariance function will be regular enough in this sense, so this assumption is not restrictive. The generalized Gaussian field over $L^2(M)$ corresponding to $f$ will be the operator $\mathrm{F}_f(g) = \langle f, g \rangle_{L^2(M)}$ for which

$$\mathbb{E}(\mathrm{F}_f(h)\,\mathrm{F}_f(g)) = \mathbb{E}\left( \langle f, h \rangle_{L^2(M)} \langle f, g \rangle_{L^2(M)} \right) = \mathbb{E} \int_M \int_M f(x)h(x)f(y)g(y)\mathrm{d}x\mathrm{d}y \qquad (94)$$

$$= \int_M \int_M \mathbb{E}(f(x)f(y))h(x)g(y)\mathrm{d}x\mathrm{d}y = \int_M \int_M k(x,y)h(x)g(y)\mathrm{d}x\mathrm{d}y = \langle Kh, g \rangle_{L^2(M)}, \qquad (95)$$

where $K : L^2(M) \to L^2(M)$ is an operator defined by $(Kh)(x) = \int_M k(x,y)h(y)dy$. Note that $\mathcal{W}$ is much less regular and *cannot* be represented this way.

Now we are ready to introduce the formal meaning of the SPDEs.

**Definition 14** (Definition 4.2.1 of Lototsky and Rozovsky [26])**.** *Let $H$ be a Hilbert space and let $\mathcal{L} : H \to L^2(M)$ be a bounded linear operator. The zero-mean generalized Gaussian random field $\mathrm{F}$ over $H$ is a solution of the equation*

$$\mathcal{L}\,\mathrm{F} = \mathcal{W} \qquad (96)$$

*if for every $g \in L^2(M)$*

$$\mathrm{F}(\mathcal{L}^* g) = \mathcal{W}(g). \qquad (97)$$

**Theorem 15** (Theorem 4.2.2 of Lototsky and Rozovsky [26])**.** *If $\mathcal{L}$ from definition 14 is invertible, then a zero-mean generalized Gaussian field $\mathrm{F}$ over $H$ defined by*

$$\mathrm{F}(h) = \mathcal{W}\left( \left(\mathcal{L}^{-1}\right)^* h \right) \qquad (98)$$

*is the unique solution of the equation* (96).

Informally, this means that $\mathrm{F} = \mathcal{L}^{-1}W$ is the solution of $\mathcal{L}\,\mathrm{F} = W$. The operator $\mathcal{L}^{-1}I\mathcal{L}^{-1} = \mathcal{L}^{-2}$ is the covariance operator of $\mathrm{F}$, which is an integral operator with some kernel $k$, which in its turn is the covariance function of $\mathrm{F}$ when viewed as an ordinary Gaussian process over the manifold $M$. The kernel $k$ is easily derived from formulas (90) and (91)—in the following, we will rigorously arrive at this result.

First, we need to introduce appropriate spaces $H$ to make $\mathcal{L} : H \to L^2(M)$ into a bounded linear bijection.

To better fit our presentation into the existing mathematical framework, we would like the operator (90) to have $2\nu/\kappa^2 = 1$. The next statement shows that this assumption does not lead to any loss of generality.

**Proposition 16.** *Consider a manifold $(M, \tilde{g})$ with $\tilde{g} = \frac{2\nu}{\kappa^2}g$, then for $\mathrm{F}$ and $\mathrm{G}$ satisfying*

$$\left( \frac{2\nu}{\kappa^2} - \Delta_g \right)^{\frac{\nu}{2}+\frac{d}{4}} \mathrm{F} = \mathcal{W}, \qquad\qquad (1 - \Delta_{\tilde{g}})^{\frac{\nu}{2}+\frac{d}{4}} \mathrm{G} = \mathcal{W}_{\tilde{g}}, \qquad (99)$$

*it is true that $\mathrm{F} = \left( \frac{\kappa^2}{2\nu} \right)^{\frac{\nu+d}{2}} \mathrm{G}$.*

We postpone the proof until after we have introduced the remaining formalism. For the time being, we assume $2\nu/\kappa^2 = 1$ when dealing with operator (90).

We proceed to define Sobolev spaces on $M$ which will serve as an appropriate $H$ for the operator (90).

**Definition 17.** *Consider $s \in (0, +\infty)$. Define the operator $(1 - \Delta_g)^{-\frac{s}{2}}$ via (88). We say that a distribution $f \in \mathcal{D}'(M)$ belongs to the Sobolev space $H^s(M)$ if and only if there exists $g \in L^2(M)$ such that $f = (1 - \Delta_g)^{-\frac{s}{2}} g$. We define the norm with $\|f\|_{H^s} = \|g\|_{L^2(M)}$, and the inner product with $\langle f, h \rangle_{H^s(M)} = \langle g, u \rangle_{L^2(M)}$, if $h = (1 - \Delta_g)^{-\frac{s}{2}} u \in H^s(M)$.*

This is one of several equivalent definition of Sobolev spaces on Riemannian manifolds, other definitions could be found in De Vito et al. [9, Theorem 3] along with a proof of their equivalence. It can be seen, thanks to our assumption $2\nu/\kappa^2 = 1$, that these spaces are particularly suitable domains for this operator 90, because these spaces are image of the inverse operator acting on $L^2(M)$.

In addition, following De Vito et al. [9], we introduce *diffusion spaces*, which will be suitable for (91).

**Definition 18.** *Consider $t \in (0, +\infty)$. Define operator $e^{\frac{t}{2}\Delta_g}$ via (88). We say that a distribution $f \in \mathcal{D}'(M)$ belongs to the diffusion space $\mathcal{H}^t(M)$ if and only if there exists $g \in L^2(M)$ such that $f = e^{\frac{t}{2}\Delta_g} g$. We define the norm with $\|f\|_{\mathcal{H}^t} = \|g\|_{L^2(M)}$ and the inner product with $\langle f, h \rangle_{\mathcal{H}^t(M)} = \langle g, u \rangle_{L^2(M)}$, if $h = e^{\frac{t}{2}\Delta_g} u \in \mathcal{H}^t(M)$.*

Both of these types of spaces are Hilbert spaces [9]. This gives the following.

**Theorem 19.** *The operators*

$$(1 - \Delta_g)^{\frac{\nu}{2}+\frac{d}{4}} : H^{\nu+\frac{d}{2}} \to L^2(M) \qquad\qquad e^{-\frac{\kappa^2}{4}\Delta} : \mathcal{H}^{\frac{\kappa^2}{2}} \to L^2(M) \qquad (100)$$

*are bounded and invertible.*

*Proof.* Immediate by definition of $H^{\nu+\frac{d}{2}}$ and $\mathcal{H}^{\frac{\kappa^2}{2}}$. $\qquad\qquad\qquad\qquad\qquad\qquad\qquad\square$

Now, we suppose that $\mathcal{L}$ is one of the operators from (100) and $H$ is the corresponding space such that $\mathcal{L} : H \to L^2(M)$. Since the conditions of Theorem 15 are satisfied, the solution of (96) is a zero-mean generalized Gaussian field F defined by (98). We now compute the covariance operator of F, which is

$$\mathbb{E}(F(h)\,F(g)) = \mathbb{E}\left(\mathcal{W}\left((\mathcal{L}^{-1})^* h\right)\mathcal{W}\left((\mathcal{L}^{-1})^* g\right)\right) = \left\langle (\mathcal{L}^{-1})^* h, (\mathcal{L}^{-1})^* g \right\rangle_{L^2(M)}, \qquad (101)$$

and since $\langle a, b \rangle_H = \langle \mathcal{L}a, \mathcal{L}b \rangle_{L^2(M)}$ is clear from definitions 17 and 18, we have for every $h \in H$ and $u \in L^2(M)$

$$\left\langle (\mathcal{L}^{-1})^* h, u \right\rangle_{L^2(M)} = \langle h, \mathcal{L}^{-1} u \rangle_H = \langle \mathcal{L}h, \mathcal{L}\mathcal{L}^{-1} u \rangle_{L^2(M)} = \langle \mathcal{L}h, u \rangle_{L^2(M)}. \qquad (102)$$

This means that $(\mathcal{L}^{-1})^* = \mathcal{L}$ and thus

$$\mathbb{E}(F(h)\,F(g)) = \left\langle (\mathcal{L}^{-1})^* h, (\mathcal{L}^{-1})^* g \right\rangle_{L^2(M)} = \langle \mathcal{L}h, \mathcal{L}g \rangle_{L^2(M)} = \langle h, g \rangle_H, \qquad (103)$$

so F is a Gaussian white noise over $H$.

We now want to obtain a Gaussian process indexed by $M$ from the generalized Gaussian field F. That is, we want to define $F(x)$ for $x \in M$ and to compute covariance function of such F. This can be easily done thanks to the fact that $H$ is a reproducing kernel Hilbert space, which was proven in De Vito et al. [9, Theorem 8, Proposition 2]—note that for the Sobolev spaces $H^s$ under consideration we always have $s > d/2$ since $s = \nu + d/2$, $\nu > 0$.

Let $k(x, x')$ be the reproducing kernel of $H$. It is natural to define $F(x) = F(k(x, \cdot))$ for $x \in M$. This $F(x)$ will be a Gaussian random variable by Definition 13. Moreover,

$$\mathbb{E}(F(x)\,F(x')) = \langle k(x, \cdot), k(x', \cdot) \rangle_H = k(x, x') \qquad (104)$$

by the definition of a reproducing kernel. It follows that $\{\mathrm{F}(x)\}_{x \in M}$ is a Gaussian process in the standard sense with zero mean and covariance function $k$ which is the reproducing kernel of $H$.[5]

The reproducing kernels for Sobolev spaces are given in De Vito et al. [9, Proposition 2] as

$$k(x, x') = \sum_{n=0}^{\infty} (1 + \lambda_n)^{-\nu - \frac{d}{2}} f_n(x) f_n(x'). \tag{105}$$

An analogous statement is true for the Diffusion spaces, giving

$$k(x, x') = \sum_{n=0}^{\infty} e^{-\frac{\kappa^2}{2} \lambda_n} f_n(x) f_n(x') \tag{106}$$

with the proof repeating the proof of [9, Proposition 2] mutatis mutandis.

Thus, the kernels normalized to have average variance $\sigma^2$ are given by

$$k_\nu(x, x') = \frac{\sigma^2}{C_\nu} \sum_{n=0}^{\infty} (1 + \lambda_n)^{-\nu - \frac{d}{2}} f_n(x) f_n(x') \tag{107}$$

$$k_\infty(x, x') = \frac{\sigma^2}{C_\infty} \sum_{n=0}^{\infty} e^{-\frac{\kappa^2}{2} \lambda_n} f_n(x) f_n(x'), \tag{108}$$

where the constant $C_{(\cdot)}$ is chosen so that $\mathrm{vol}_g(M)^{-1} \int k_{(\cdot)}(x, x) \mathrm{d}x = \sigma^2$. In some cases, for instance when $M$ is a homogeneous manifold, $k_{(\cdot)}(x, x)$ will not depend on $x$, so $k(x, x) = \sigma^2$ can be satisfied.[6]

Note that throughout the above, we still assumed $\kappa$ is chosen such that $2\nu/\kappa^2 = 1$. To show this assumption was indeed taken without loss of generality, we prove the following.

**Proposition 20.** *Consider a manifold* $(M, \tilde{g})$ *with* $\tilde{g} = \frac{2\nu}{\kappa^2} g$, *then for* F *and* G *satisfying*

$$\left(\frac{2\nu}{\kappa^2} - \Delta_g\right)^{\frac{\nu}{2} + \frac{d}{4}} \mathrm{F} = \mathcal{W}, \qquad\qquad (1 - \Delta_{\tilde{g}})^{\frac{\nu}{2} + \frac{d}{4}} \mathrm{G} = \mathcal{W}_{\tilde{g}}, \tag{99}$$

*it is true that* $\mathrm{F} = \left(\frac{\kappa^2}{2\nu}\right)^{\frac{\nu+d}{2}} \mathrm{G}$.

*Proof.* First, let us verify that the equation to the left is well-defined. To do this, we must check that operator (90) is bounded and invertible for general $\kappa, \nu > 0$. Fix $f \in H^{\nu + \frac{d}{2}}$ and find $g \in L^2(M)$ such that $f = (1 - \Delta_g)^{-\frac{\nu}{2} - \frac{d}{4}} g$. Write $g = \sum_{n=0}^{\infty} \alpha_n f_n$ using the basis $\{f_n\}$ consisting of Laplacian eigenfunctions, so $f = \sum_{n=0}^{\infty} (1 + \lambda_n)^{-\frac{\nu}{2} - \frac{d}{4}} \alpha_n f_n$. Noting that

$$\min\left(\frac{2\nu}{\kappa^2}, 1\right) \le \frac{\frac{2\nu}{\kappa^2} + \lambda_n}{1 + \lambda_n} \le \max\left(1, \frac{2\nu}{\kappa^2}\right) \tag{109}$$

we can write

$$\left\|\left(\frac{2\nu}{\kappa^2} - \Delta_g\right)^{\frac{\nu}{2} + \frac{d}{4}} f\right\|_{L^2(M)}^2 = \left\|\sum_{n=0}^{\infty} \left(\frac{2\nu/\kappa^2 + \lambda_n}{1 + \lambda_n}\right)^{\frac{\nu}{2} + \frac{d}{4}} \alpha_n f_n\right\|_{L^2(M)}^2 \tag{110}$$

$$= \sum_{n=0}^{\infty} \left(\frac{2\nu/\kappa^2 + \lambda_n}{1 + \lambda_n}\right)^{\nu + \frac{d}{2}} \alpha_n^2 \le \sum_{n=0}^{\infty} \max\left(1, \frac{2\nu}{\kappa^2}\right)^{\nu + \frac{d}{2}} \alpha_n^2 \tag{111}$$

$$= \max\left(1, \frac{2\nu}{\kappa^2}\right)^{\nu + \frac{d}{2}} \|g\|_{L^2(M)}^2 = \max\left(1, \frac{2\nu}{\kappa^2}\right)^{\nu + \frac{d}{2}} \|f\|_{H^{\nu + \frac{d}{2}}}^2, \tag{112}$$

which proves boundedness as well as $f \in D\left(\left(\frac{2\nu}{\kappa^2} - \Delta_g\right)^{\frac{\nu}{2} + \frac{d}{4}}\right)$. To prove the operator is invertible, write

$$\left\|\left(\frac{2\nu}{\kappa^2} - \Delta_g\right)^{\frac{\nu}{2} + \frac{d}{4}} f\right\|_{L^2(M)}^2 = \sum_{n=0}^{\infty} \left(\frac{2\nu/\kappa^2 + \lambda_n}{1 + \lambda_n}\right)^{\nu + \frac{d}{2}} \alpha_n^2 \tag{113}$$

$$\geq \min\left(\frac{2\nu}{\kappa^2}, 1\right)^{\nu + \frac{d}{2}} \sum_{n=0}^{\infty} \alpha_n^2 = \min\left(\frac{2\nu}{\kappa^2}, 1\right)^{\nu + \frac{d}{2}} \|f\|_{H^{\nu + \frac{d}{2}}}^2. \tag{114}$$

Now, consider how a change of the metric from $g$ to $\tilde{g} = \frac{2\nu}{\kappa^2} g$ changes the objects under consideration. This is given by the standard expressions

$$\Delta_{\tilde{g}} = \frac{\kappa^2}{2\nu} \Delta_g, \qquad \widetilde{dx} = \left(\frac{2\nu}{\kappa^2}\right)^{d/2} dx, \tag{115}$$

which in turn gives

$$\tilde{\lambda}_n = \frac{\kappa^2}{2\nu} \lambda_n, \quad \tilde{f}_n = \left(\frac{2\nu}{\kappa^2}\right)^{-d/4} f_n, \quad \langle f, g \rangle_{\tilde{g}} = \left(\frac{2\nu}{\kappa^2}\right)^{d/2} \langle f, g \rangle, \quad \mathcal{W}_{\tilde{g}} = \left(\frac{2\nu}{\kappa^2}\right)^{d/4} \mathcal{W}. \tag{116}$$

With this, we have

$$(1 - \Delta_{\tilde{g}})^{\frac{\nu}{2} + \frac{d}{4}} G = \sum_{n=0}^{\infty} \left(1 + \tilde{\lambda}_n\right)^{\frac{\nu}{2} + \frac{d}{4}} \left\langle G, \tilde{f}_n \right\rangle_{\tilde{g}} \tilde{f}_n = \sum_{n=0}^{\infty} \left(1 + \frac{\kappa^2}{2\nu} \lambda_n\right)^{\frac{\nu}{2} + \frac{d}{4}} \langle G, f_n \rangle f_n \tag{117}$$

$$= \left(\frac{\kappa^2}{2\nu}\right)^{\frac{\nu}{2} + \frac{d}{4}} \sum_{n=0}^{\infty} \left(\frac{2\nu}{\kappa^2} + \lambda_n\right)^{\frac{\nu}{2} + \frac{d}{4}} \langle G, f_n \rangle f_n = \left(\frac{\kappa^2}{2\nu}\right)^{\frac{\nu}{2} + \frac{d}{4}} \left(\frac{2\nu}{\kappa^2} - \Delta_g\right)^{\frac{\nu}{2} + \frac{d}{4}} G. \tag{118}$$

This means that G is a solution of

$$\left(\frac{\kappa^2}{2\nu}\right)^{\frac{\nu}{2} + \frac{d}{4}} \left(\frac{2\nu}{\kappa^2} - \Delta_g\right)^{\frac{\nu}{2} + \frac{d}{4}} G = \left(\frac{2\nu}{\kappa^2}\right)^{d/4} \mathcal{W}. \tag{119}$$

Gathering all constants, we get that $F = \left(\frac{\kappa^2}{2\nu}\right)^{\frac{\nu}{2} + \frac{d}{4}} \left(\frac{2\nu}{\kappa^2}\right)^{-d/4} G = \left(\frac{\kappa^2}{2\nu}\right)^{\frac{\nu+d}{2}} G$ is the solution to

$$\left(\frac{2\nu}{\kappa^2} - \Delta_g\right)^{\frac{\nu}{2} + \frac{d}{4}} F = \mathcal{W} \tag{120}$$

which proves the statement. $\qquad \square$

This means that the kernel of a Gaussian process solving $\left(\frac{2\nu}{\kappa^2} - \Delta_g\right)^{\frac{\nu}{2} + \frac{d}{4}} F = \mathcal{W}$ is proportional to

$$k(x, x') = \sum_{n=0}^{\infty} \left(1 + \tilde{\lambda}_n\right)^{-\nu - \frac{d}{2}} \tilde{f}_n(x) \tilde{f}_n(x') = \left(\frac{\kappa^2}{2\nu}\right)^{-\nu - d} \sum_{n=0}^{\infty} \left(\frac{2\nu}{\kappa^2} + \lambda_n\right)^{-\nu - \frac{d}{2}} f_n(x) f_n(x'). \tag{121}$$

Re-normalizing this kernel, we finally get

$$k_\nu(x, x') = \frac{\sigma^2}{C_\nu} \sum_{n=0}^{\infty} \left(\frac{2\nu}{\kappa^2} + \lambda_n\right)^{-\nu - \frac{d}{2}} f_n(x) f_n(x'), \tag{122}$$

where $C_\nu$ is chosen as above and $\kappa$ can now be any positive number. Together with

$$k_\infty(x, x') = \frac{\sigma^2}{C_\infty} \sum_{n=0}^{\infty} e^{-\frac{\kappa^2}{2} \lambda_n} f_n(x) f_n(x') \tag{123}$$

given in (108), this gives the kernels we sought, and concludes our presentation.

## Footnotes

[4]Such a manifold is automatically complete, since a compact metric space is always complete.

[5]It is easy to see that $\mathcal{B}_{\mathrm{F}}(g) := \langle \mathrm{F}, g \rangle_H$, where F is the Gaussian process on $M$, is the generalized Gaussian field F we started with.

[6]It is not known to the authors if homogeneous manifolds are the only manifolds for which $k(x, x)$ does not depend on $x$. It seems like an interesting mathematical problem to describe manifolds with this property. It is even more interesting to describe how the way $k(x, x)$ changes depending on $x$ is determined by the geometry of $M$.