[Reviews · NeurIPS 2020]

Review 1

Summary and Contributions: The authors propose a method that enables computing the Matern and the squared exponential kernel on compact Riemannian manifolds without boundary. Using the eigenbasis of the Laplace-Beltrami operator, they derive the kernel function. The effectiveness of the methodology is demonstrated using some simple examples. ===== After rebuttal ===== I would like to thank the authors for their responses. My main concern is the readability of the paper. Since the authors agree to do some improvements to make the camera-ready version more accessible, I am happy to raise my score and vote for weak acceptance.

Strengths: The problem that the papers aims to solve is quite important and interesting, since the naive squared exponential kernel under the geodesic distance is not in general positive definite on a Riemannian manifold. The mathematical analysis seems to be solid.

Weaknesses: I find the paper very hard to read and to understand. The mathematical content is too much and not so easily accessible from a non-expert reader.

Correctness: The technical part of the submission seems to be correct. However, I find it very hard to verify the technical part of the paper, especially in such a short time.

Clarity: I think that the paper is very hard to read for the average machine learning researcher. In my opinion, the manuscript can be improved such that to be more accessible to machine learning audience.

Relation to Prior Work: I think that the relation to prior work is discussed to some extend. Probably, a more clear comparison between the Lindgren et al. would have been beneficial.

Reproducibility: Yes

Additional Feedback: I am pretty sure that this is a good idea, and also, that the technical part is correct. However, as a non-expert, I find it very hard to read and to understand all the details of the paper. There is a lot of mathematical content compressed. I am not sure how accessible is the current version of the manuscript from the average machine learning researcher. Therefore, I am not sure about the impact that could have in the community.


Review 2

Summary and Contributions: This paper aims to adapt the GP regression model, and in particular some of the tools used in such models, beyond Euclidean domains. It relies on periodic summation to provide expressions for the kernel of a Matérn GP that extends it to general compact Riemannian manifolds without boundary.

Strengths: This work addresses an interesting and important problem of using GP regression on Riemannian manifolds. The theoretical grounding of the work appears sound and the results that are provided illustrate the claims reasonably well.

Weaknesses: I find the novelty and the significance of the work somewhat hard to assess. As far as I understand, the paper proposes a technique for performing GP regression with Matern and SE kernels on Riemannian manifolds. However, the paper does not include an explicit previous work section on this topic, and the main reference used in the introduction is that to Lindgren et al. [23]. Lindgren et al. [23] proposes defining a Matern GP as a solution to a particular SPDE, and this submission extends the approach of Lindgren et al. [23]. However, my understanding is that GP regression on Riemannian manifolds could be described in other ways, too (see later question on previous work). As an extension to Lindgren et al. [23], this work offers some merits; in particular, it provides a constructive definition and allows for use of other standard GP tools, such as sparse approximations. However, the authors don't seem to provide any further comparisons to Lindgren et al. [23]. My understanding (and please correct me if I'm wrong) is that this work would improve the stability and the scalability of Lindgren et al. [23]. However, there are no direct comparisons refering to that. It might be informative to provide a comparison of the computational complexity of the proposed approach and [23]. Also, in lines 35-36 it sounds as if the proposed approach does not require using an FEM solver. However, my understand is that both [23] and the proposed approach use an FEM solver. It might be good to provide a more detailed explanation of how the proposed approach leads to a more stable behaviour of the FEM solver.

Correctness: The main ideas and results provided in the main part of the paper seemed correct to me. However, I have not checked the details of the proofs provided in the appendix.

Clarity: The quality of the writing is good overall. My main concern is that, if I understand correctly, the paper is meant to be aimed at applications, and with that in mind, some parts of it are a little hard to follow (though I appreciate that the work relies on a some complex machinery, and so it is not possible to explain all of it in detail in 8 pages).

Relation to Prior Work: As mentioned above, some comparison to previous work would be interesting. See for example, [1] A. Mallasto, A. Feragen. Wrapped Gaussian Process Regression on Riemannian Manifolds. (CVPR), 2018 (and the references in it)

Reproducibility: Yes

Additional Feedback: My understand is that one of the limitations of [23] is the restriction of the smoothness of the Matérn parameter. Would it be instructive to demonstrate empirically how your proposed approach overcomes these limitations, or is there some reason why it might not be possible? Is it true that the computational complexity of [23] is actually lower than in your case (my understanding is that they use a basis function representation with piece-wise linear basis functions which they do for efficiency)? In general, the idea and the potential applications of this work are very interesting. My main concern is that it is not clear to me where this work fits in the broader context, and how applicable it is in practical terms (for example, in terms of computational cost or easy of use). --------------------------------AFTER REBUTTAL-------------------------------------------- I thank the authors for addressing my questions. I think it would be helpful to add the comments on the relation to Lindgren et al. [23] to the final version to make it easier for the readers (especially those who are not experts in the area) to see the advantages of your approach.


Review 3

Summary and Contributions: The paper is purely theoretic. It considers Gaussian processes (GPs) with the Matern kernel on compact Riemannian manifolds without boundary. A general expression of this kernel is provided in terms of a spectral decomposition. This is a potentially useful result as previous approaches have relied on stochastic PDEs on manifolds, which is rather impractical. == After the rebuttal == The rebuttal was convincing, so I stand by my score of acceptance. I agree with one of my fellow reviewers that the material is not particularly accessible outside a small subset of the NeurIPS community. This may not by easy to "fix" but I encourage the authors to listen to this feedback and try to adapt.

Strengths: The paper brings forward a new theoretical analysis for generalizing GPs to manifolds. The work provides a path towards being able to evaluate the associated kernel, which has generally not been possible so far. As such, this is important work.

Weaknesses: I have two concerns about the paper. (1) The work is only applicable to compact manifolds without boundary. In this case, the generalization from Euclidean space is essentially handled by wrapping the process around the manifold. It is not clear to me how one might go about extending these results to the non-compact case, which is highly important. (2) It is unclear to me how easy it is to actually apply the theory in practice. It seems that one will need access to the eigenfunctions of the Laplace-Beltrami operator, which seems just as difficult to get as the kernel itself. Say, for example, that I work on a manifold with known Exp/Log maps, can I then use the theory or do I need more analytical knowledge in order to derive a suitable kernel?

Correctness: As far as I can tell, the work is indeed correct.

Clarity: The paper is generally well-written. I would have preferred if title and abstract had accurately reflected that the work only applies to compact manifolds without boundary, as a superficial reading of the paper will give the incorrect impression that the theory applies to all Riemannian manifolds (it does not).

Relation to Prior Work: The authors generally do a good job of positioning the paper. One relevant line of related work has, however, been missed and should be acknowledged: Wrapped Gaussian Process Regression on Riemannian Manifolds A. Mallasto and A. Feragen CVPR - IEEE Conference on Computer Vision and Pattern Recognition 2018 This paper defines GPs on manifolds through a wrapping process, which is quite similar to what is done in the present paper. The ideas are only vaguely related so, I do not think this paper diminishes the contribution of the present paper. It might also be relevant to mention the follow-up work Probabilistic Riemannian submanifold learning with wrapped Gaussian process latent variable models A. Mallasto, S. Hauberg and A. Feragen AISTATS - International Conference on Artificial Intelligence and Statistics 2019 which extend the first paper to arrive at the GP-LVM on manifolds.

Reproducibility: Yes

Additional Feedback: I was confused in lines 55-56 where it is states that henceforth GP priors are assumed to have zero mean: what does that mean on a manifold? I understand the given intuition in line 142-150 of why the naive generalizations break-down, but this is strongly linked to the compactness of the considered manifolds. The issue with the naive generalization is more general, so I fear that the stated intuition might actually lead to confusion in the general case (where the stated intuition does not apply). I think it would be good to be more explicit about this intuition being strongly linked to compactness, and therefore constitute a simplification of the general problem.


Review 4

Summary and Contributions: The literature on Gaussian process models has primarily been focused on the traditional setting where a problem is defined on the Euclidean space. On the other hand, problems defined on other spaces such as Riemannian manifolds often require an orthogonal set of tools and techniques for enabling the application of GPs to that setting. In this work, the authors are concerned with bridging the gap between these two directions of work by showing how kernels from the family of the Matern covariance function can be reliably constructed for data living on such manifolds. Whereas previous attempts at doing so were either complex to integrate into standard GP libraries (due to the requirement for solving SPDEs) or poorly specified (for example the naive extension to incorporate geodesic distances in a covariance function), the approach developed here is generalisable to Riemannian manifolds where no boundary is set, while the formulation of the kernel makes it possible to the borrow from other improvements related to GPs which have been developed for problems in the Euclidean space (e.g. random feature approximations, inducing point approximations, online learning, etc).

Strengths: As noted in my paper summary, the problem setting considered in this paper is often overlooked within the GP community, which is why I believe that the paper’s contributions could be very helpful for reigniting interest in this class of problems. Beyond just having theoretical significance, I also agree with the authors that translating the formulation of such methods in a manner which makes them amenable to the usual GP tools and tricks could ‘demystify’ the use of such methods for many practitioners, who would otherwise have to resort to working with SPDE solvers in order to apply such models in practice. I look forward to seeing this work integrated in a widely-used GP library such as GPflow or GPyTorch. I also complement the authors for clearly demonstrating how a naive attempt extending standard GP kernels to the geodesic setting may only work in very limited settings. Although this issue was indeed identified in earlier work, I appreciated the way in which the authors framed their subsequent contributions in light of this more ‘immediate’ approach.

Weaknesses: Although the paper’s contributions are primarily of a theoretic nature, putting more effort into constructing a more complete real-world example could be helpful for giving more context to which problem settings are amenable to such methods. For example, since the authors highlight climate sciences as being one of several domains where one could expect to operate in the manifold setting, having such an example could definitely enhance the paper’s accessibility. While the current synthetic examples do a good job in showcasing the construction’s robustness when applied to complex problems, readers and practitioners who are unfamiliar with this class of problems may find it difficult to assimilate the examples to a more standard problem setting.

Correctness: Although I am very familiar with work on Gaussian processes, I admit to being less acquainted with how these are adapted to problems beyond the Euclidean space. However, all contributions presented in this work appear to be throughly backed up via proofs in the supplement and specific pointers to related work in this area. It should be possible to reproduce all of the paper’s main results using the details provided across the main paper and supplement. On an unrelated note, throughout the paper, the authors often mention that the approximation developed here is sensible as long as the truncation includes a certain number of terms. I feel as though this comment could perhaps be expanded upon further in an updated version of the paper, perhaps via including an additional synthetic example or carrying out some ablation study on the truncation order.

Clarity: The paper is very well-written and was a pleasure to read. The authors do a good job in preserving the essential details regarding their contributions in the main text, while deferring lengthier proofs and derivations to the supplementary material. I thank the authors for properly proof-reading the paper before submitting - only spotted one tiny typo in L24: ‘application areas’.

Relation to Prior Work: The scope of this work in relation to the literature on similar models is adequately presented in the paper.

Reproducibility: Yes

Additional Feedback: - - - Comments on rebuttal - - - Thank you for your rebuttal. I appreciated all the points touched upon in your response, and trust that the suggested improvements and clarifications in regards to related work and applications will be properly incorporated in a prospective camera-ready version of the paper.

[Author Response · NeurIPS 2020]

**R1**, **R2**, **R3**, **R4**: First, we would like to thank the reviewers for their time, effort, and helpful reviews! There were some common concerns and minor misunderstandings about the scope of the paper and context of previous work. We will use the additional space of the camera-ready version to elaborate on these points, which we now address.

• **Recap: what is our goal?** To connect the SPDE characterization of Riemannian Matérn GPs with familiar tools of the GP community, e.g. inducing point approximations, Fourier feature methods, stochastic variational inference.

• **Recap: how do we achieve this?** By introducing explicit expressions (18, 19) to compute Matérn kernels on a manifold point-wise, and explicit expressions (20, 21) for Fourier features approximations.

• **Recap: what are the prerequisites?** One needs to know the eigenfunctions and eigenvalues of the Laplace–Beltrami operator on the manifold of interest. For many cases these are analytic, but can also be obtained numerically.

• **Recap: what do we mean by "Gaussian processes on manifolds"?** We refer to GPs whose **inputs** lie on a manifold, and whose outputs lie in $\mathbb{R}$ (or $\mathbb{R}^d$) as usual, i.e. they are random functions $f : M \to \mathbb{R}$.

**R1**, **R2**, **R3**, **R4**: **writing, clarity, & compression of mathematical content.** We are grateful that most referees thought our work was well-written: we tried hard to present everything as accessibly as possible given the technical nature of the topic. We will incorporate referees' suggestions in the final version to improve the presentation further.

**R1**, **R2**, **R3**: **differences with Lindgren et al. (especially when a finite element (FEM) solver is required).**

• **Zero/one solve vs. multiple solves**. Our approach does not always require a FEM solve, because for many manifolds (spheres, tori, Stiefel manifolds, symmetric spaces, Grassmannian manifolds, and many others) Laplace–Beltrami eigenpairs have known analytic expressions. If these are not analytic, then in our approach a FEM solve needs to be performed once in advance as a precomputation, which is well-studied and can be done in a controlled fashion with high accuracy. Lindgren et al. require running a FEM solver at training time, with all the ensuing consequences.

• **Complexity and cost of solve.** In cases where FEM solves are required, the cost and complexity of both our method and Lindgren et al. will depend primarily on the interplay between the order of the (S)PDE to be solved and dimension/geometry of the manifold. For Lindgren et al., higher smoothness values will necessitate higher-order FEM (usually piecewise polynomial) spaces. In contrast, Laplace–Beltrami eigenpairs are obtained by solving a second-order PDE, where a piecewise linear FEM space suffices. Since piecewise linear FEM spaces tend to be less expensive, we expect in most cases that cost and complexity of our method will be favorable compared to Lindgren et al.

• **Hyperparameters and gradients.** To compute the gradient of hyperparameters, Lindgren et al. require one to solve an adjoint PDE at training time. Using our method, these are computed straightforwardly via automatic differentiation.

• **Restrictions on smoothness and ease of use.** Lindgren et al. require the smoothness $\nu$ to be chosen so that the resulting SPDE is of integer order. We impose no such restrictions: $k_\nu(x, x') = \frac{\sigma^2}{C_\nu} \sum_{n=0}^{\infty} \left( \frac{2\nu}{\kappa^2} + \lambda_n \right)^{-\nu - \frac{d}{2}} f_n(x) f_n(x')$ depends on $\nu$ in a simple arithmetical manner. Since computation of $\lambda_n$ and $f_n$ up to some truncation level is standard functionality in most FEM packages, our method is easy to use and requires less code or FEM expertise.

**R2**, **R3**: **the "Wrapped Gaussian Process Regression on Riemannian" citation and "manifold zero mean" comment.** That citation considers (generalized) Gaussian processes with manifold **outputs**, not inputs, i.e. functions $g : \mathbb{R} \to M$. This makes it completely technically different from our work ($f : M \to \mathbb{R}$). However, we recognize some readers might be looking for this case, so we will add a citation and explicitly state that this is **not** what we are doing.

**R3**: **compactness.** We completely agree that the focus on the compact case should be stated explicitly in the abstract, and will add this. The cylinder considered in Section 5.1 is not compact: this case is possible because it is the product of a (compact) circle and a line. We expect our method can be generalized to many interesting non-compact cases, such as tangent bundles of compact manifolds and other constructions used in physics. We expect the general non-compact case to require substantially heavier technical machinery, such as spectral decompositions via projection-valued measures.

**R3**, **R4**: **generality of technique and applied use cases.** Laplace–Beltrami eigenfunctions are a widely-used technical tool for working with manifolds: to calculate these numerically, an embedding into $\mathbb{R}^d$ suffices. Others such as Ye et al. (Biometrika 2020, arXiv:2006.14266) have studied Riemannian kernels based on explicit $\exp/\log$ maps. We find it valuable to have different techniques available with their own requirements to suit practitioners' needs. The dynamical systems example illustrates a simplified use case in robotics, where GPs are used for data-efficient learning. Similarly, Jaquier et al. (CoRL 2019, arXiv:1910.04998) could benefit from using our machinery instead of the ill-defined naïve generalization. We also expect use cases in climate science, such as modeling of sea surface temperatures on earth.

**R3**: **intuition in lines 142-150.** The naïve generalization can indeed be formulated more generally on geodesic spaces without any manifold structure. Unfortunately, Feragen et al. have a similar no-go theorem in this setting, which says that these kernels are only well-defined for geodesic spaces that are flat in the sense of Alexandrov. In our view, the intuition in Section 3 is linked more with the Abelian Lie group structure, rather than compactness per se: this is discussed in **lines 162-165**. It does indeed break down in more general scenarios, but we still consider it helpful.

**R4**: **truncation.** This is an important point: truncation error will depend on the kernel's smoothness parameter. In Section 5.2, we used $500$ eigenpairs without perceptible accuracy issues. We will include additional discussion on this.

[Meta-Review · NeurIPS 2020]

The four reviewers, all of whom are seasoned domain experts, consistently agree that this paper is conceptually solid, timely and relevant. I should thus be accepted. The reviewers also pointed out some issues with the presentation. I want to strongly encourage the authors to take these into account to increase the potential audience of this paper.